# A GABAergic Maf-expressing interneuron subset regulates the speed of locomotion in *Drosophila*

H. Babski[1]*, T. Jovanic[2,3], C. Surel[1], S. Yoshikawa[4,7], M. F Zwart [5], J. Valmier[1], J.B. Thomas[4], J. Enriquez[6], P. Carroll [1,8] & A. Garcès[6,8]*

Interneurons (INs) coordinate motoneuron activity to generate appropriate patterns of muscle contractions, providing animals with the ability to adjust their body posture and to move over a range of speeds. In *Drosophila* larvae several IN subtypes have been morphologically described and their function well documented. However, the general lack of molecular characterization of those INs prevents the identification of evolutionary counterparts in other animals, limiting our understanding of the principles underlying neuronal circuit organization and function. Here we characterize a restricted subset of neurons in the nerve cord expressing the Maf transcription factor Traffic Jam (TJ). We found that TJ$^+$ neurons are highly diverse and selective activation of these different subtypes disrupts larval body posture and induces specific locomotor behaviors. Finally, we show that a small subset of TJ$^+$ GABAergic INs, singled out by the expression of a unique transcription factors code, controls larval crawling speed.

[1] Inserm U1051, Institute for Neurosciences of Montpellier, University of Montpellier, 34295 Montpellier, France. [2] Paris-Saclay Institute of Neuroscience, UMR 9717, 1 avenue de la Terrasse, 91190 Gif-sur-Yvette, France. [3] Decision and Bayesian Computation, Pasteur Institute, CNRS UMR 3571, Paris, France. [4] Molecular Neurobiology Laboratory, The Salk Institute for Biological Studies, La Jolla, CA 92037, USA. [5] School of Psychology and Neuroscience, University of St Andrews, St Andrews KY16 9JP, UK. [6] Institut de Génomique Fonctionnelle de Lyon (IGFL), École Normale Supérieure de Lyon and Centre National de la Recherche Scientifique (CNRS), 32 avenue Tony Garnier, 69007 Lyon, France. [7] Present address: Department of Molecular, Cellular, and Developmental Biology, University of California Santa Barbara, Santa Barbara, CA 93106, USA. [8] These authors contributed equally: P. Carroll, and A. Garcès,  *email: helene.babski@inserm.fr; alain.garces@inserm.fr

The wiring and functioning of the neuronal circuits that control locomotion have been extensively studied[1]. In mammals, as in invertebrates, the speed of locomotion is regulated by central pattern generator (CPG) neuronal circuits which coordinate, via motoneurons (MNs), the sequential activation of muscles[2–5]. The rhythmic bursts of MN activity are controlled by local excitatory and inhibitory interneurons (INs) located in the ventral region of the vertebrate spinal cord and in the ventral nerve cord (VNC) of invertebrates. These INs can be grouped into subclasses on the basis of their connectivity patterns, physiological properties such as neurotransmitter used, and by the set of specific transcription factors (TFs) they express. The diversity of INs is best exemplified by a recent study showing that the V1 IN class in mammals can be fractionated into 50 distinct subsets on the basis of the expression of 19 TFs[6]. By capitalizing on the combinatorial expression of TFs, the function of highly restricted subpopulations of INs can be assessed. For example, using transgenic lines and intersectional genetics, one can precisely control the expression of ion channel proteins and thus regulate IN activity[7,8]. Such an approach has proved instrumental in dissecting the core logic of the CPG circuits that generate the rhythm and pattern of motor output in both vertebrates and in Drosophila[3,5,9]. Yet despite the remarkable diversity of INs in the vertebrate spinal cord[6,10] and in the Drosophila VNC[11,12], and the resultant genetic tools to manipulate small subsets of cells, a thorough description of the functioning of the CPG regulating locomotion in animals is far from complete.

A class of segmentally arrayed local premotor inhibitory INs named PMSIs (for period-positive (Per+) median segmental INs) has recently been found to control the speed of larval locomotion by limiting, via inhibition, the duration of MN output. The PMSIs have been proposed to be the fly equivalent of the V1 INs in the mouse. Thus PMSIs in Drosophila and V1 INs in vertebrates may represent a phylogenetically conserved IN population that shapes motor output during locomotion[5]. In the years following the characterization of the PMSIs, specific IN subtypes that contribute to the diversity of locomotor behaviors in the Drosophila larvae have been identified, providing a wealth of information on each IN subpopulation, their function, morphology, and synaptic connections[5,13–15]. However, little is known about the combinatorial expression of TFs within these different IN subtypes; this lack of knowledge impedes cross-species comparisons, thus limiting our understanding of the common principles of CPG organization in vertebrates and invertebrates.

Here we characterize in the Drosophila nerve cord a small pool of highly diverse INs (23/hemisegment) expressing the evolutionarily conserved TF Traffic Jam (TJ), the orthologue of MafA, MafB, c-Maf, and NRL in the mouse. Interestingly, like TJ in Drosophila, MafA, MafB, and c-Maf are expressed in the developing mouse spinal cord by restricted subpopulations of ventral premotor INs[6,10,16]. To characterize the TJ-expressing INs we generated a TJ-Flippase line and used intersectional genetics to activate specific TJ+ subpopulations on the basis of their neurotransmitter properties. We found that manipulation of these IN subsets modulates larval locomotor behaviors in specific ways. Activation of a restricted subpopulation of GABAergic/Per+/TJ+ neurons (3 INs/segment), belonging to the PMSIs group of INs and known as MNB progeny neurons, significantly impacts the crawling speed of the larvae. Finally, we provide evidence that these MNB progeny neurons belong to the recently characterized Ladder neurons, a subtype of feedforward inhibitory INs that mediate behavioral choice during mechanosensory responses in the larva [17].

## Results

**TJ-expressing neurons are involved in larval locomotion.** We initiated a detailed analysis of the TF TJ expression in the embryonic and larval nervous systems, using a previously characterized TJ-specific antibody[18] and an enhancer trap for TJ (TJ-Gal4)[19] (Table 1). During embryogenesis, TJ expression is first detected in late stage 12 (st 12) in 12 to 15 cells/hemisegment in the VNC and in few cells in the brain (Supplementary Fig. 1a, Fig. 2j). Co-immunostaining with the glial marker Repo showed that TJ is excluded from glia cells (Supplementary Fig. 1b). We found that TJ-Gal4 faithfully recapitulates TJ expression in all embryonic (Supplementary Fig. 1c) and larval stages analyzed (Fig. 1c–g, Supplementary Fig. 1g). Detailed analysis of TJ expression over time showed that TJ is consistently found in a subset of 29 neurons per hemisegment in the VNC abdominal region (A2–A6) from embryonic st17 to L3 larval stages (Fig. 1a, b, Supplementary Fig. 1d–g, Supplementary Movie 1). We used TJ-Gal4::UAS-H2AGFP in combination with anti-TJ immunostainings to establish a precise topographic map of TJ+ neurons in second instar larvae, a stage representative of the stable expression pattern of TJ throughout development (Fig. 1c–g).

Next, we explored the function of TJ+ neurons in larval locomotion using the TJ-Gal4 driver as a tool to either silence or activate the entire TJ+ neuronal population. We silenced neurons by expressing thermosensitive shibire (shi^ts)[20]; neuronal activation was achieved by expressing TrpA1[21]. Silencing of the entire TJ+ population led to a slight, but statistically significant, decrease in the number of larval peristaltic waves (Fig. 1h, second beige squares). Larvae display general disorganization of the peristaltic waves, with the segments of the animal failing to contract in a coordinated and sequential manner (Fig. 1j, k, Supplementary Movie 2). Interestingly, activation of the TJ+ neurons had more drastic effects, with larvae exhibiting almost complete abolition of locomotion (Fig. 1i, second red squares) and some exhibiting a complete paralysis which we named "spastic paralysis". This phenotype was characterized by immobility, tonic contraction of all body segments and a drastic shortening of the whole larval body length (Fig. 1l, Supplementary Movie 3). When placed back at permissive temperature (23 °C), the larvae resumed normal locomotion, proving that this spastic paralysis phenotype is fully reversible. TJ is expressed in a restricted number of neurons in the brain hemispheres. To assess the role of these neurons in the spastic paralysis phenotype, we restricted the expression of TrpA1 to the TJ+ brain neurons using Tsh-Gal80[22] in combination with TJ-Gal4 and UAS-TrpA1. Under these conditions locomotion appeared normal (Fig. 1i, second salmon pink squares), arguing that TJ+ neurons in the brain hemispheres do not function prominently in Drosophila larval locomotion.

We thus conclude that the activity of TJ+ neurons within the VNC are important for Drosophila larval crawling and that the normal function of at least some of the TJ+ neurons is to maintain proper muscle contraction and peristaltic wave propagation during locomotion.

**Activation of TJ+ neurons in the VNC induces paralysis.** To further characterize the identity of TJ+ neurons regulating crawling behavior in the larva we developed an intersectional-based genetic approach, using candidate LexA drivers to express a LexAop>Stop>dTrpA1 transgene (>denotes FRT site; generous gift from Y. Aso, Janelia Research Campus) in combination with a source of Flippase. To specifically target TJ+ neurons we generated a TJ-Flippase line. Briefly, within a ~40 kb backbone fosmid carrying all the endogenous regulatory elements of tj, we substituted the full open reading frame and 3′ UTR sequences of tj with an optimized Flippase (FlpO) by in vivo homologous recombination in Escherichia coli[23] (see Methods for more details). We first characterized the accuracy and efficiency of

**Table 1 List of antibodies, Drosophila lines, oligonucleotides and reagents used in this study**

| Primary antibodies | Species | Dilution | Source |
| --- | --- | --- | --- |
| **Antibodies list** | | | |
| Anti-TJ | Guinea Pig | 1:4000 | Gift from D. Godt[18] |
| Anti-Engrailed 4D9 | Mouse | 1:50 | Development Studies Hybridoma Bank, University of Iowa |
| Anti-pMad | Rabbit | 1:500 | [45] |
| Anti-GFP | Chicken | 1:2000 | Abcam Ab13970 |
| Anti-GFP | Rabbit | 1:4000 | Invitrogen A6455 |
| Anti-RFP | Rabbit | 1:8000 | Gift from S. Heidmann[46] |
| Anti-myc 9E10 | Mouse | 1:40 | Development Studies Hybridoma Bank, University of Iowa |
| Anti-eve 3C10 | Mouse | 1:40 | Development Studies Hybridoma Bank, University of Iowa |
| Anti-eve | Rabbit | 1:200 | Gift from M. Frasch[47] |
| Anti-vGlut (C-ter) | Rabbit | 1:1000 | Gift from H. Aberle[48] |
| Anti-Gad1 818 | Rabbit | 1:500 | Gift from F.R. Jackson[49] |
| Anti-Repo 8D12 | Mouse | 1:40 | Development Studies Hybridoma Bank, University of Iowa |
| Anti-Prospero | Mouse | 1:40 | Development Studies Hybridoma Bank, University of Iowa |
| Anti-FasII 1D4 | Mouse | 1:80 | Development Studies Hybridoma Bank, University of Iowa |
| Anti-5-HT | Rabbit | 1:2000 | Immunotech ref. 0601 |
| Anti-Jumu | Rabbit | 1:800 | Gift from J. Enriquez[50] |

| Secondary antibodies | Species | Dilution | Source |
| --- | --- | --- | --- |
| Anti-guinea pig A488 | Goat | 1:1000 | Alexa Fluor™ (Invitrogen) |
| Anti-guinea pig A555 | Goat | 1:2000 | |
| Anti-guinea pig A647 | Donkey | 1:1000 | |
| Anti-chicken A488 | Goat | 1:1000 | |
| Anti-mouse A488 | Donkey | 1:1000 | |
| Anti-mouse A405 | Goat | 1:1000 | |
| Anti-mouse A555 | Donkey | 1:2000 | |
| Anti-mouse A647 | Donkey | 1:1000 | |
| Anti-rabbit A488 | Donkey | 1:1000 | |
| Anti-rabbit A555 | Donkey | 1:2000 | |
| Anti-rabbit A647 | Donkey | 1:1000 | |
| **Other reagents** | | | |
| Phalloïdin-FluoProbes 647 | NA | 1:25 | FP-BA0320, Interchim |

| *Drosophila* lines | Source | | Identifier |
| --- | --- | --- | --- |
| ***Drosophila* stocks** | | | |
| *TJ-Gal4* | Tokyo Stock Center | | DGRC # 104055 |
| *TJ-Flp* | Generated by the lab | | NA |
| *Isl$^{-Tmyc}$* | [27] | | |
| *Gad1-LexA* | Bloomington Stock Center | | BL # 60324 |
| *vGlut-LexA* | Bloomington Stock Center | | BL # 60314 |
| *ChAT-LexA* | Bloomington Stock Center | | BL # 60319 |
| *UAS-TrpA1* | Bloomington Stock Center | | BL # 4308 |
| *UAS-shi$^{ts}$* | [20] | | NA |
| *lexAop > stop > dTrpA1* (2 lines on II$^d$ and III$^d$ chromosomes) | Obtained from the Rubin lab (Janelia Research Campus) | | NA |
| *Per-LexA* | Tokyo Stock Center | | DGRC # 116999 |
| *Per-Gal4* | Bloomington Stock Center | | BL # 7127 |
| *FkhGFP* | Bloomington Stock Center | | BL # 43951 |
| *Tsh-LexA* | Gift from J. Simpson (UC Santa Barbara)[41] | | NA |
| *Tsh-Gal80* | Gift from G. Miesenböck (University of Oxford)[22] | | NA |
| *lexAop-IVS-tdTomato.nls* | Bloomington Stock Center | | BL # 66680 |
| *UAS-H2AGFP* | [51] | | NA |
| *20XUAS-6XGFP* | Bloomington Stock Center | | BL # 52262 |
| *Act»Gal4* | Bloomington Stock Center | | BL # 4779 |
| *UAS-CD8GFP* | Bloomington Stock Center | | BL # 5137 |
| *CQ2-LexA* | Gift from C. Doe (University of Oregon)[13] | | NA |
| *sim-Gal4* (or *sim3.7-Gal4*) | Bloomington Stock Center | | BL # 9150 |
| *Hlh3bGFP* | Gift from P. Tomancak (Max Planck Institute) (unpublished) | | NA |
| *Grain-lacZ* | [52] | | NA |
| *Gad1$^{AD}$* | Bloomington Stock Center | | BL # 60322 |
| *UAS-LexA$^{DBD}$* | Bloomington Stock Center | | BL # 56528 |
| *UAS-myrGFP* | Bloomington Stock Center | | BL # 32198 |
| *B-H1-Gal4* | [53,54] | | NA |
| *JRC-SS00863* (split Gal4 for Ladder-d) | [17] | | |
| *Tdc2-Gal4* | Bloomington Stock Center | | BL # 9313 |
| *Dimm-Gal4* | Bloomington Stock Center | | BL # 25373 |
| *TH-Gal4* (also called *ple-Gal4*) | Bloomington Stock Center | | BL # 8848 |
| *vmatGFP* | Bloomington Stock Center | | BL # 60263 |
| *UAS»TrpA1$^{myc}$* | Bloomington Stock Center | | BL # 66871 |
| *lexAop-Flp* | Bloomington Stock Center | | BL # 55820 |
| *lexAop > stop > CsChrimson$^{Venus}$* | Obtained from Rubin lab (Janelia Research Campus) | | NA |
| *GMR47G08-LexA* | Bloomington Stock Center | | BL # 52793 |

| Components | Plasmid of origin | | Source |
| --- | --- | --- | --- |
| **Generation of the *TJ-Flippase*** | | | |
| kanR | 2xTY1-kanR | | Gift from P. Tomancak (Max Planck Institute)[23] |
| Late SV40 pA | pCAGGS-Ires2-eGFP-linkerPacl-sv40pA | | Gift from J.F. Brunet (ENS Paris) |
| nlsFlpO | pPGK FlpO bpA | | Gift from S. Bourane, Goulding lab (Salk Institute) |
| pFly-fosmid-TJ | NA | | Gift from P. Tomancak[23] |
| pRed-Flp4 | NA | | |

| Oligonucleotide name | Sequence | Use |
| --- | --- | --- |
| recTJsens | 5'ATGTGAGACCCGTAATCGACCCTC TCCGGTCCCTGGTCGATCCAATGAA A**ATGGCTCCTAAGAAGAAG<u>AGG</u>** (TJ 5' homologous sequence) **(FlpO 5' sequence)** ATG: start codons in *tj* | Amplification of FlpO-lateSV40pA-kanR cassette to add homologous recombination arms to it |
| recTJantisens | 5'ATTCATTAATTTAGATTTATTTAT TACTAAATTGTTTTATGCACACTTA T**AACTCAGAAGAACTCGTCAAG** (TJ 3' homologous sequence) **(KanR 3' sequence)** | Amplification of FlpO-lateSV40pA-kanR sequence to add homologous recombination arms to it |

Oligonucleotides were ordered from Eurofins Genomics

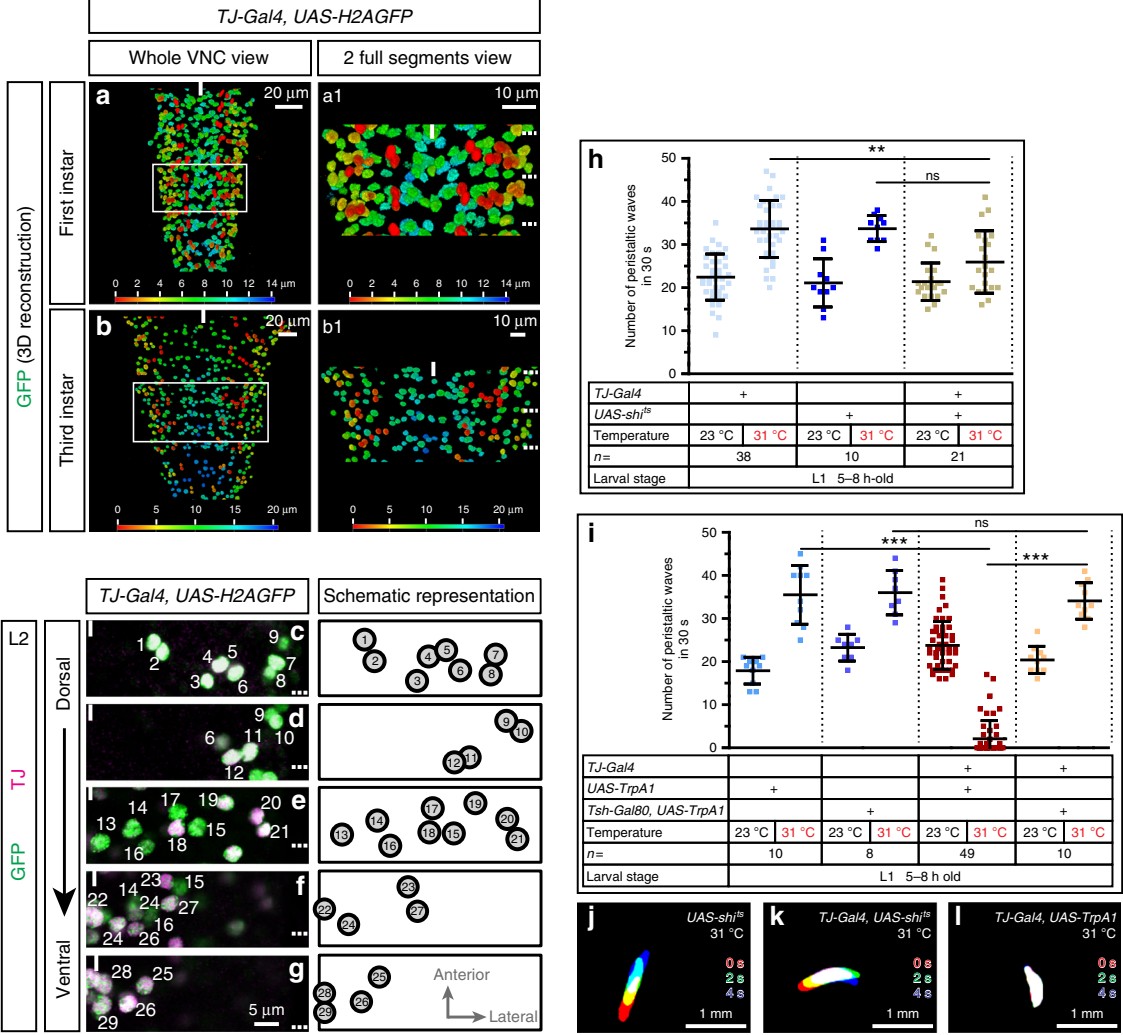

**Fig. 1** TJ+ neurons are required for proper larval crawling. **a**, **b** 3D reconstruction of whole VNC of first (**a**) and third (**b**) instar larvae expressing nuclear GFP under the control *TJ-Gal4*. Color scale is the *z*-axis scale; most dorsal cells are red, and most ventral are blue. White scale is the *x/y*-axis scale. **c**–**g** Staining of a second instar larva VNC for TJ (magenta) and *TJ-Gal4* expression reported by nuclear H2AGFP (green). Totality of TJ-expressing cells are shown in dorsal (**c**) to ventral (**g**) panels. Dashed lines on the right-hand side of the panels indicate segment boundaries and the full line the midline. A unique hemisegment is shown in each panel. Anterior of the VNC is up. Right panels are schematic representations of one hemisegment showing stereotyped ventral–dorsal and medial–lateral cell position of TJ-expressing cells. **h**, **i** Number of peristaltic waves per 30 s at (23 °C) and (31 °C). Note that larvae naturally increase the numbers of peristaltic waves at 31 °C compared to 23 °C. Silencing of the entire TJ+ population (second beige dot plot, **h**) causes a slight decrease in the number of peristaltic waves. Activation of the entire TJ+ population (second red dot plot) causes a drastic decrease in the number of peristaltic waves (**i**). This decrease is no longer visible upon activation of the TJ+ neurons exclusively in the brain (second salmon pink dot plot, **i**). For **h** and **i**, each single point represents recording of a single first instar larva. Error bars indicate the mean ± SD and *n* the number of larvae tested. Statistical analysis: Graph **i**: one-way ANOVA. ***$p \leq 0.001$, ns = not significantly different. Graph **h**: Kruskall–Wallis. **$p \leq 0.01$, ns = not significant. **j**–**l** Superimposition of three consecutive time frames (0, 2, and 4 s) showing the postures of 5-h-old larvae: control larvae (**j**), upon silencing of the TJ+ neuronal population (**k**) and upon activation of the TJ+ population (**l**)

*TJ-Flp* in "flip-out" experiments using *Act>Stop>Gal4::UAS-CD8-GFP*. We found that 64.5% of the TJ-expressing neurons have recombined in young first instar (L1) larvae (Fig. 2a–c). By early L2, recombination has reached 89% (Fig. 2d–f). Using an alternative "flip-out" approach (*TJ-Flp* in combination with *LexAop>Stop>Chrimson-Venus::Tsh-LexA*) we found that 74.5% of TJ-expressing neurons in young L1 larvae have recombined (Supplementary Fig. 2d–f). Thus, recombination triggered by *TJ-Flp* is both accurate and efficient. Further confirming the accuracy of *TJ-Flp*, an examination of developing egg chambers of the ovary, a structure in which TJ has been reported to be specifically expressed by somatic cells[18], revealed that all TJ+ follicular and border cells have recombined (Supplementary Fig. 2b, c). Finally, we used *Tsh-LexA* to express a *LexAop>Stop>dTrpA1*

transgene in combination with *TJ-Flp*, anticipating that activation of TJ+ neurons in the VNC only should give rise to the drastic spastic paralysis phenotype described above. Indeed, we found that peristaltic waves were completely abolished, thus preventing crawling behavior, in almost all L1 larvae tested (*n* = 12) (Fig. 2k, Supplementary Movie 4) and in third instar larvae (Fig. 2l).

Collectively, these results show that the *TJ-Flp* line we generated is an accurate and powerful tool to manipulate TJ+ neurons, thus validating our intersectional-based genetic approach.

**Activation of TJ+ MNs has modest effect on crawling.** We next asked whether the spastic paralysis phenotype we observed upon

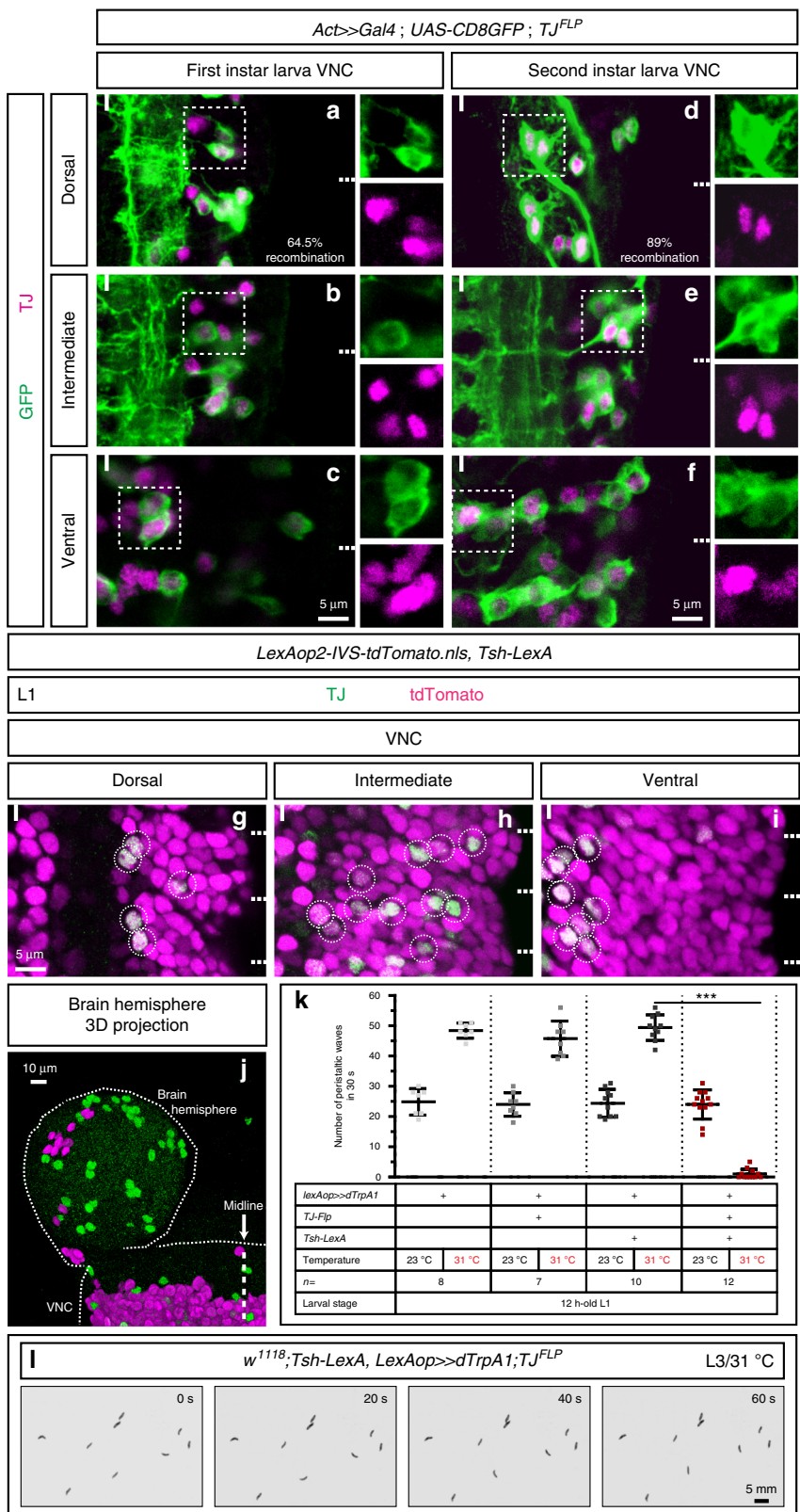

**Fig. 2** *TJ-Flp* accurately targets TJ⁺ neuronal subpopulations. **a–f** Staining for TJ (magenta) and recombined cells (expressing *TJ-Flp* –green) in 0–6 h old first instar (**a–c**) and young second instar (**d–f**) larva VNC. Percentages of cells recombined are indicated. **g–j** Staining for TJ (green) and *Tsh-LexA* driving an *nls-tdTomato* (magenta) in first instar larva VNC (**g–i**) and in first instar larva brain hemisphere (**j**). All TJ⁺ cells in the VNC are *Tsh-LexA*⁺ while no co-localization is found in the brain. **j** A single brain hemisphere and the anterior part of the VNC are visible (left side). **k** Number of peristaltic waves per 30 s at (23 °C) and (31 °C) temperatures. Each single point represents a single 12-h-old first instar larva. Error bars indicate the mean ± SD and *n* denotes the number of larvae tested. Statistical analysis: Kruskall–Wallis test ***p < 0.001. **l** Snapshots extracted from a 60 s time-lapse recording of nine L3 larvae upon activation of all TJ⁺ neurons located in the VNC using the MWT (Multi-Worm Tracker) automated behavioral device

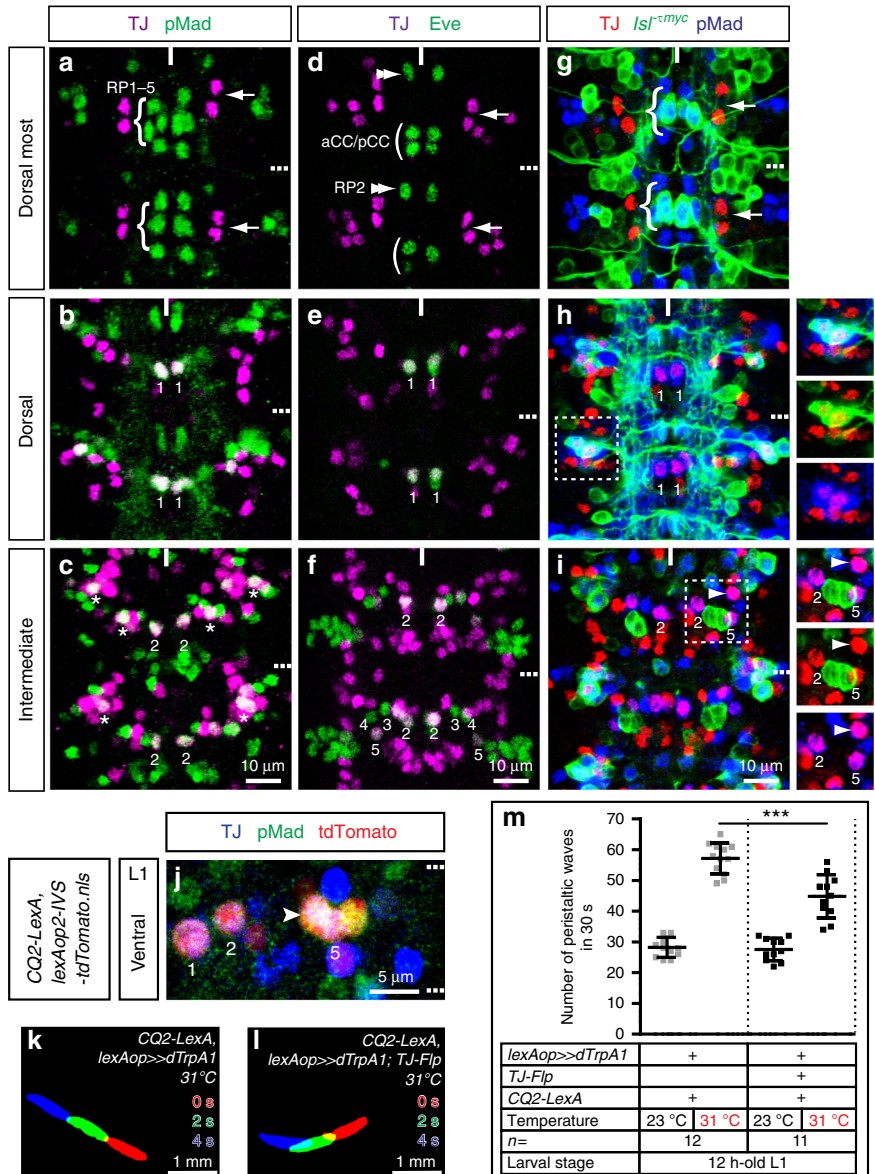

**Fig. 3** TJ+ motoneuron activation moderately impairs larval locomotion. **a–i** Representative views of two segments of stage 16 embryonic VNC. **a–c** are images from a single VNC and the same applies to **d–f** and **g–i**. **a–c** TJ (magenta) and the motoneuron (MN) marker Phospho Mad (pMad) (green). **d–f** TJ (magenta) and Eve (green). **g–i** TJ (red), pMad (blue), and *Isl−Tmyc* (green). TJ is excluded from RP1–5 (**a** curly brackets) and expressed in CQ/U MNs (**b**, **c** 1 for U1 and 2 for U2; other TJ+/pMad+ MNs cannot be precisely identified and are labeled with asterisks). Arrows in **a** highlight the dorsal-most pair of TJ+ neurons. TJ is excluded from aCC/pCC (**d** parenthesis) and RP2 (**d** double arrowhead) and expressed in CQ/U MNs (**e**, **f** 1 for U1, 2 for U2, 3 for U3, 4 for U4, and 5 for U5). Arrows in **d** highlight the dorsal-most pair of TJ+ neurons. TJ is expressed in two dorso-lateral *Isl*+/pMad+ MNs (**h**, inset), in the *Isl*- CQ/U MNs (**h**, **i** U1, U2, and U5) and in 1 TJ+/pMad+ MN (**i**, full arrowhead). Arrows in **g** highlight the dorsal-most pair of TJ+ neurons and numbers 1, 2, and 5 in **h** and **i** show U1, U2, and U5 MNs, respectively. **j** Staining for TJ (blue), pMad (green), and endogenous *nls-tdTomato* driven by *CQ2-LexA* in a first instar larva VNC. *CQ2-LexA* drives in TJ+ CQ/U MNs U1 (numbered 1), U2 (numbered 2), and U5 (numbered 5) as well as in an extra TJ+/pMad+ MN (full arrowhead). A full hemisegment is shown. Midline is to the left. **k**, **l** Superimposition of three consecutive time frames (0, 2, and 4 s) showing the postures of 12-h-old larvae: control larvae (**k**) and upon activation of part of the TJ+ MNs population (**l**). **m** Number of peristaltic waves per 30 s at (23 °C) and (31 °C) temperatures. Each single point represents a single 12-h-old first instar larva. Error bars indicate the mean ± SD and *n* denotes the number of larvae tested. Statistical analysis: one-way ANOVA. ***$p \leq 0.001$

activation of the entire TJ+ population might actually be caused by *TJ-Gal4* expression in MNs. Using phosphorylated Mad (pMad) and Even-skipped (Eve) as reliable molecular markers for MNs[24–26] we found TJ expression from embryonic st13 onward in three CQ/U MNs, namely U1, U2, and U5 (respectively labeled 1, 2, and 5 in Fig. 3b, c, e, f, h, i), plus another putative pMad+ MN located in the vicinity of the CQ/U MNs (full arrowhead in Fig. 3i). Co-labeling using *Isl-Tmyc*[27] and pMad also revealed TJ expression in two MNs located dorsolaterally in the VNC that are

putatively part of the ISNd MN pool (Fig. 3h, inset). We noticed that TJ was not expressed in aCC, in any RP MNs (RP1–5) (Fig. 3a, d, g), in SNa MNs (Supplementary Fig. 3a, b), nor in type II, octopaminergic *Tdc2*+ MNs (Supplementary Fig. 3c).

To confirm and further delineate the identity of TJ+ MNs we used *TJ-Gal4* to express a myristoylated targeted RFP reporter (*UAS-myr-RFP*) (generous gift from M. Landgraf). From late embryonic st16 to L3, *TJ-Gal4::UAS-myr-RFP* revealed the peripheral projections as well as the terminal processes of the

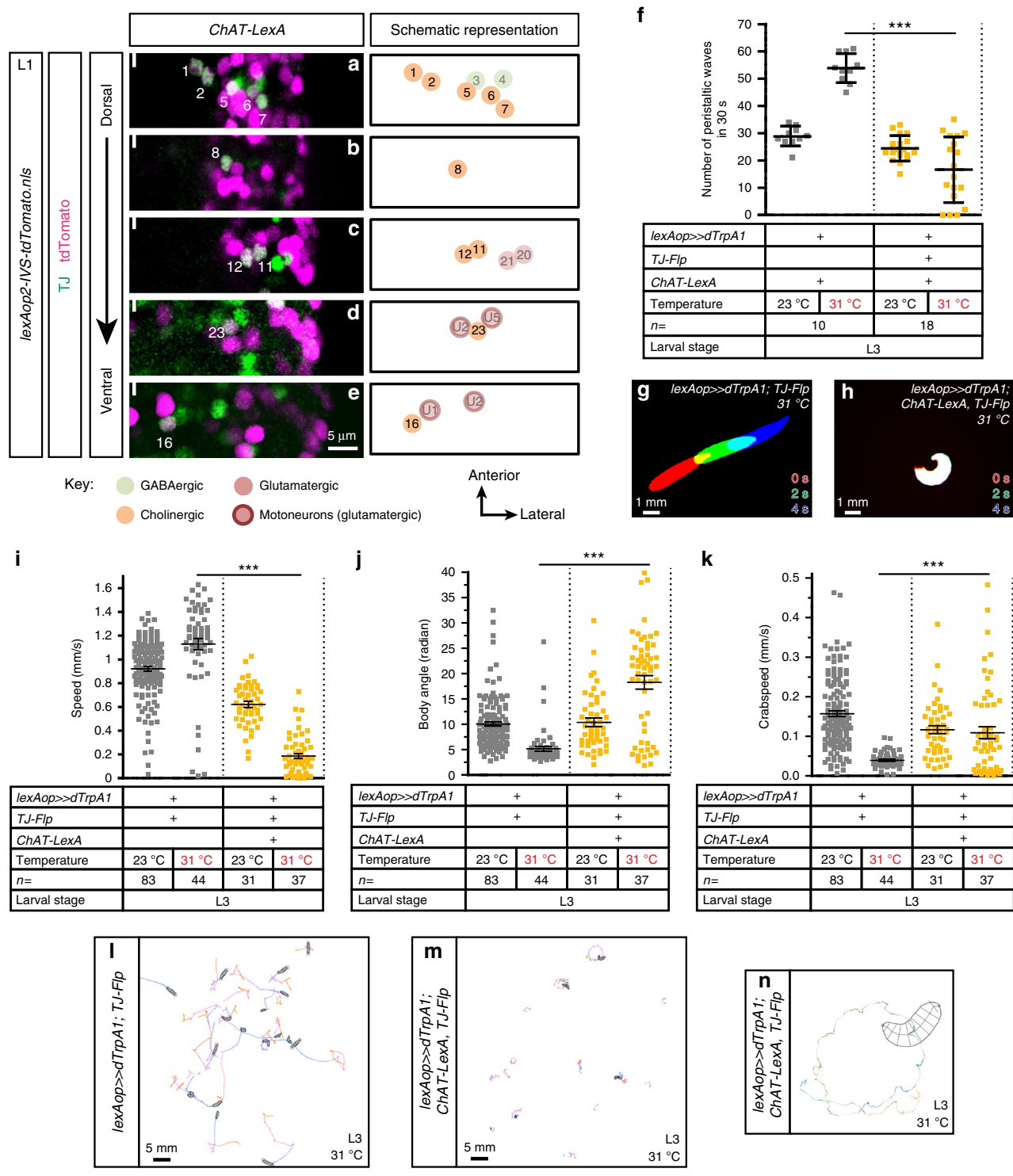

Key:
- GABAergic
- Glutamatergic
- Cholinergic
- Motoneurons (glutamatergic)

| | lexAop>>dTrpA1 | | | |
|---|---|---|---|---|
| **f** | + | | + | |
| TJ-Flp | | | + | |
| ChAT-LexA | | | + | |
| Temperature | 23 °C | 31 °C | 23 °C | 31 °C |
| n= | 10 | | 18 | |
| Larval stage | L3 | | | |

| | lexAop>>dTrpA1 | | | |
|---|---|---|---|---|
| **i** | + | | + | |
| TJ-Flp | + | | + | |
| ChAT-LexA | | | + | |
| Temperature | 23 °C | 31 °C | 23 °C | 31 °C |
| n= | 83 | 44 | 31 | 37 |
| Larval stage | L3 | | | |

| | lexAop>>dTrpA1 | | | |
|---|---|---|---|---|
| **j** | + | | + | |
| TJ-Flp | + | | + | |
| ChAT-LexA | | | + | |
| Temperature | 23 °C | 31 °C | 23 °C | 31 °C |
| n= | 83 | 44 | 31 | 37 |
| Larval stage | L3 | | | |

| | lexAop>>dTrpA1 | | | |
|---|---|---|---|---|
| **k** | + | | + | |
| TJ-Flp | + | | + | |
| ChAT-LexA | | | + | |
| Temperature | 23 °C | 31 °C | 23 °C | 31 °C |
| n= | 83 | 44 | 31 | 37 |
| Larval stage | L3 | | | |

TJ+ MNs on muscles VO3–VO6 (Supplementary Fig. 4a–c) and on muscles LL1, DO5, DO2, and DO1 (Supplementary Fig. 4d–f). These results are in agreement with the above immunostaining results. We conclude that in each abdominal hemisegment there is a contingent of two TJ+/pMad+/Islet$^{tmyc+}$ MNs (ISNd MNs that project to muscles VO3–VO6), three TJ+/pMad+/Eve+ MNs (ISNdm MNs U1, U2 and U5, that project to muscles DO1, D02 and LL1, respectively), and 1 TJ+/pMad+ MN that projects to muscle DO5. To investigate the role of TJ+ MNs in locomotion we used CQ2-LexA[13], which drives expression in the three TJ+

CQ/U MNs U1, U2, and U5. Detailed monitoring of CQ2-LexA expression revealed that in mid-L1 stage larvae this line also drives expression in an additional TJ+ MN (arrowhead in Fig. 3j). Thus, CQ2-LexA drives expression in four of the six TJ+ MNs per hemisegment. We used CQ2-LexA to activate these neurons and observed only a moderate decrease in the number of peristaltic waves, with no dramatic locomotor phenotypes or body posture defects in the vast majority of the larvae (Fig. 3m). Importantly, in contrast to activation of all TJ+ neurons using the TJ-Gal4 driver, no spastic paralysis was observed upon activation of TJ+ MNs

**Fig. 4** Activation of TJ[+] cholinergic INs reduces crawling speed and induces ventral bending and rolling. **a–e** Representative views of one hemisegment of a first instar larval VNC stained for TJ (green) and cholinergic cells (magenta; using *ChAT-LexA* driving *lexAop-nlsTomato*). Cells are shown from dorsal (**a**) to ventral (**e**) positions. The full line indicates the midline and anterior of the VNC is up. TJ[+] cholinergic neurons are represented in bright color while other non-cholinergic TJ[+] neurons are shown with paler colors. The identity of those neurons is inferred from their position compared to TJ[+] cholinergic neurons. **f** Number of peristaltic waves per 30 s of larvae expressing TrpA1 in TJ[+] cholinergic neurons (orange dot plot) versus their respective controls that do not express TrpA1 (gray dot plot). Each single point represents a single third instar larva. Error bars indicate the mean ± SD and *n* the number of larvae tested. Statistical analysis: Kruskall–Wallis ***$p \leq 0.001$. **g, h** Superimposition of three consecutive time frames (0, 2, and 4 s) showing the postures of third instar control larvae (**g**) and upon activation of the TJ[+] cholinergic population (**h**). **i** Speed of larvae expressing TrpA1 in TJ[+] cholinergic neurons (orange dot plot) versus their respective controls that do not express TrpA1 (gray dot plot). **j** Body angle (in radian) of larvae expressing TrpA1 in TJ[+] cholinergic neurons (orange dot plot) versus their respective controls that do not express TrpA1 (gray dot plot). **k** Crab speed of larvae expressing TrpA1 in TJ[+] cholinergic neurons (orange dot plot) versus their respective controls that do not express TrpA1 (gray dot plot). **l–k** Statistical analysis: Z-test. ***$p \leq 0.001$.
**l, m** Traces of third instar control larvae (**l**) and third instar larvae upon activation of the TJ[+] cholinergic neurons (**m**) undergoing locomotion for 30 s. Traces were generated using the MWT (Multi-Worm Tracker) automated behavioral device. **n** Close-up of the trace done by a L3 larva upon activation of the TJ[+] cholinergic neurons in 30 s using the MWT

using the *CQ2-LexA* driver (Fig. 3k, l, compare Supplementary Movies 5 and 6).

From this experiment we conclude that activation of four out of six hemisegmental TJ[+] MNs does not trigger the spastic paralysis phenotype observed upon activation of the entire TJ[+] population, but rather causes a slight defect in locomotion, likely due to the constant contraction of the dorsal muscles innervated by TJ[+] MNs.

**Subsets of TJ[+] neurons impinge on distinct motor behaviors.** We next monitored larvae in which we activated different subpopulations of TJ-expressing INs with TrpA1. We assayed the effects by counting the number of peristaltic waves and by quantifying several larval behaviors using automated tracking software (MWT—Multi-Worm Tracker) (Supplementary Movies 7 and 8)[28,29]. We first explored the possible expression of TJ in peptidergic, dopaminergic, serotonergic, and histaminergic INs and found no expression of TJ in these IN subclasses (Supplementary Fig. 3d–j). Further molecular characterization using highly specific LexA drivers[30] showed that 10 out of 29 TJ[+] neurons per hemisegment are cholinergic (Fig. 4a–e, Supplementary Fig. 5d–f). Activation of these TJ[+] cholinergic INs by intersectional genetics disrupted normal crawling (Fig. 4f–n). Larvae frequently adopted a characteristic "crescent shape", with head and tail regions brought close together by tonic contraction of the ventral muscles (Fig. 4h, Supplementary Movie 9). This phenotype was heterogeneous in terms of severity: some larvae were continually immobile with ventral contraction (Supplementary Movie 9), while others displayed bouts of ventral contraction interrupting otherwise seemingly normal crawling. Finally, some larvae displayed a rolling behavior (Fig. 4m, n, Supplementary Movies 10 and 11). Interestingly, such a spectrum of severity has recently been found for ventral contraction phenotypes caused by activation of different subsets of VNC neurons[9]. Regardless of the heterogeneity of the phenotype, the number of peristaltic waves produced by TJ[+] cholinergic IN-activated larvae was significantly decreased compared to controls (Fig. 4f) as was the average crawling speed (Fig. 4i). Body angle was increased, as might be expected from the observed ventral contractions and rolling behaviors (Fig. 4j). The crab speed, which can be used as a proxy measure for rolling[28], was increased as well (Fig. 4k).

Five TJ[+] INs per hemisegment are glutamatergic and differentiating them from TJ[+] MNs, which are also glutamatergic, is possible by counterstaining with pMad (Fig. 5a–f, Supplementary Fig. 5g–i). We chose to use *vGlut-LexA* in combination with *TJ-Flp* to activate this IN population, keeping in mind that we would also be activating TJ[+] MNs. We found that activation of the entire TJ[+] glutamatergic contingent (5 INs and 6 MNs per hemisegment)

caused a complete paralysis of nearly all the larvae (Fig. 5h), with forward propagating waves largely absent (Fig. 5g, i). Another feature of this phenotype was the vertical lifting of the most anterior segments (thoracic head region) off the substrate (Supplementary Movie 12). Automated tracking of immobile larvae using our automated tracking device was not possible due to the inability of the program to detect static objects. However, for those "escaper" larvae that kept moving we found that the average speed was decreased (Fig. 5j) and body angle, an indicative parameter of head casting or turning, was increased as the larvae displayed repetitive head casts and head retractions (Fig. 5k, Supplementary Movie 13). The fact that the phenotype induced by activation of TJ[+] glutamatergic neurons is substantially different from the one observed upon activation of four out of six hemisegmental TJ[+] MNs (see section above), argues for the TJ[+] glutamatergic INs playing an important role in the normal locomotor behavior of the larvae.

The remaining eight TJ[+] INs per hemisegment are GABAergic (Fig. 6a–d and Supplementary Fig. 5a–c). Examining *Gad1-LexA* co-localization with TJ, we noticed that the three most ventral co-expressing neurons are located at the midline (neurons no. 22, 28, 29 in Fig. 6c, d) and do not appear to be bilaterally paired, indicating that these cells are unpaired, midline cells[31]. Activation of the TJ[+] GABAergic IN subpopulation led to seemingly normal locomotion (Fig. 6f, g and compare Supplementary Movies 14 and 15). However, there was a 37% reduction of the number of peristaltic waves compared to control specimen at 31 °C (Fig. 6e). Automated tracking of the larvae revealed that their crawling speed is decreased compared to controls at 31 °C (Fig. 6j). This reduction in average speed results from actual slower crawling bouts as the animals moved in straight and smooth lines (Fig. 6h, i and Supplementary Movie 16). Consistent with the latter observation, we found that their body angle is significantly reduced compared to control (Fig. 6k–m).

Taken together, these results show that activation of restricted subpopulations of TJ[+] neurons defined on the basis of their neurotransmitter identity impacts larval behavior differently. Cholinergic neurons induce "crescent shape" or rolling behavior, glutamatergic neurons induce paralysis, lifting of anterior body or rapid head casts and head retractions while activation of GABAergic neurons impacts the crawling speed of larvae that navigate in smooth runs.

**Midline TJ[+] GABAergic neurons impact crawling speed.** While searching for more restricted LexA drivers that would allow us to further subdivide the TJ[+] population implicated in locomotion, we identified *Per-LexA*[5] whose expression co-localizes with nine of the most ventral TJ[+] neurons per segment (Fig. 7a–f; neurons no. 22, 24, 25, 26, 28, and 29 on Fig. 1f, g). Activation of the nine

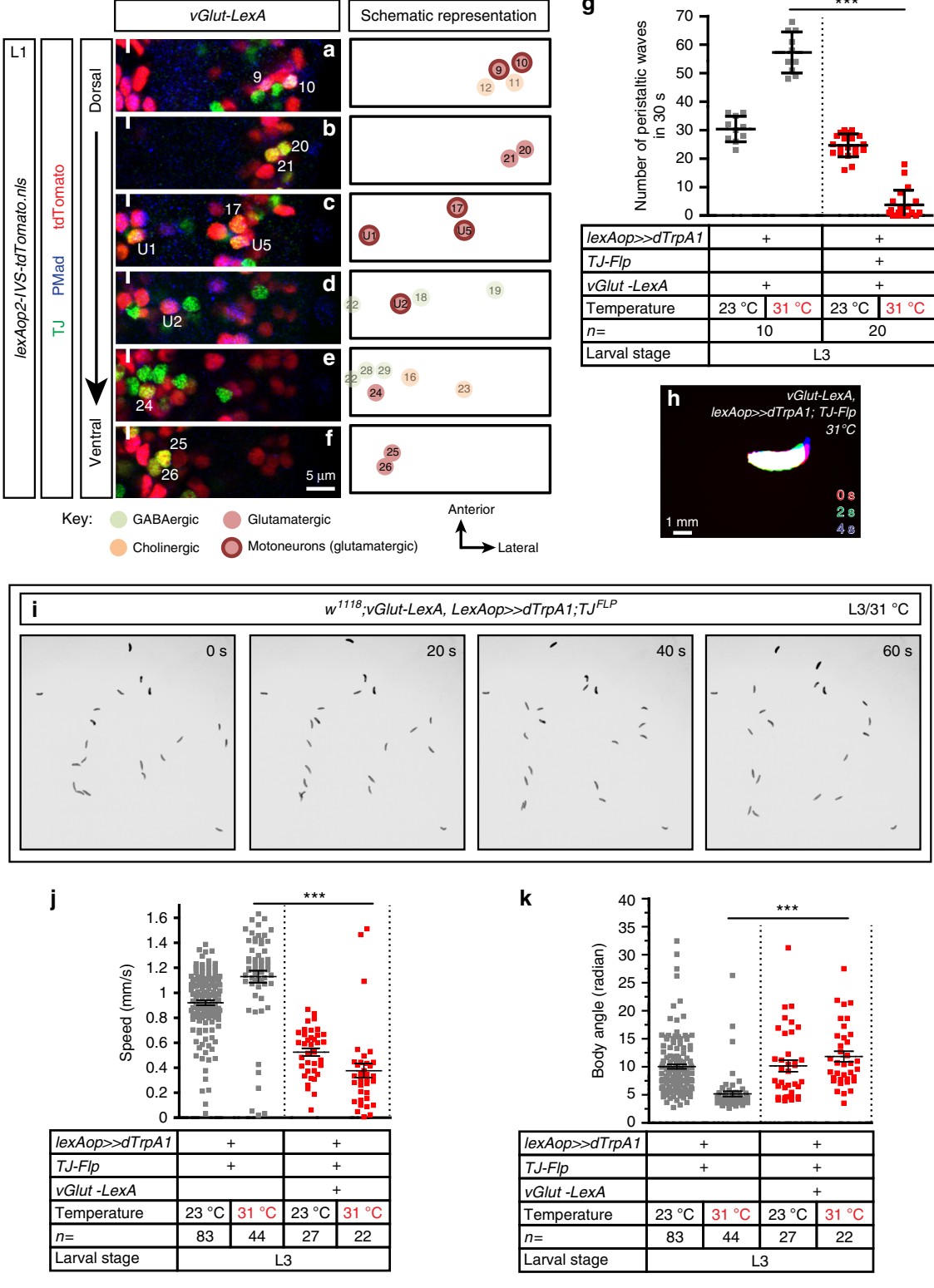

- GABAergic
- Cholinergic
- Glutamatergic
- Motoneurons (glutamatergic)

TJ$^+$/Per$^+$ neurons per segment resulted in a decrease in the number of peristaltic waves (Fig. 7g, Supplementary Movie 17). In these animals, the speed of locomotion was decreased (Fig. 7j) while body angle was increased compared to controls (Fig. 7k). Indeed, larvae frequently displayed bouts of active head casts and back-ups (that we could not quantify) interrupting otherwise seemingly normal crawling episodes (Supplementary Movie 18).

Upon further characterization of the TJ$^+$/Per$^+$ population, we discovered that six of the neurons (three per hemisegment) are glutamatergic (Fig. 7d–f; neurons no. 24, 25, 26 in Fig. 1f, g) while the remaining three, located medially, are GABAergic (Fig. 7a–c). These three TJ$^+$/Per$^+$ GABAergic neurons correspond to the three unpaired, midline GABAergic cells described above (see Fig. 6c, d; neurons no. 22, 28, 29). We used a triple intersectional

**Fig. 5** Activation of TJ[+] glutamatergic neurons impacts locomotor speed and triggers head casting. **a–f** Representative views of one hemisegment of a first instar larval VNC stained for TJ (green), glutamatergic cells (red, using *vGlut-LexA* driving *lexAop-nlsTomato*), and the motoneuron marker pMad (blue). Cells are shown from dorsal (**a**) to ventral (**f**) positions. The full line indicates the midline and anterior of the VNC is up. TJ[+] glutamatergic neurons are represented in bright color while other non-glutamatergic TJ[+] neurons are shown with paler colors. The identity of those neurons is inferred from their position compared to TJ[+] glutamatergic neurons. **g** Number of peristaltic waves per 30 s at (23 °C) and (31 °C) of larvae expressing TrpA1 in TJ[+] glutamatergic neurons (red dot plot) versus their respective controls that do not express TrpA1 (gray dot plot). Each single point represents a single third instar larva. Error bars indicate the mean ± SD and *n* the number of larvae tested. Statistical analysis: one-way ANOVA. ***$p \leq 0.001$. **h** Superimposition of three consecutive time frames (0, 2, and 4 s) showing the postures of third instar larvae upon activation of the TJ[+] glutamatergic population. **i** Time-lapse of L3 larvae upon activation of all TJ[+] glutamatergic neurons using the MWT (Multi-Worm Tracker). Note that the larvae are found in similar locations during the duration of recording indicating that they were mainly immobile. **j** Speed of larvae expressing TrpA1 in TJ[+] glutamatergic neurons (red dot plot) versus their respective controls that do not express TrpA1 (gray dot plot). **k** Body angle (in radian) of larvae expressing TrpA1 in TJ[+] glutamatergic neurons (red dot plot) versus their respective controls that do not express TrpA1 (gray dot plot). **j, k** Statistical analysis: *Z*-test. ***$p \leq 0.001$

approach to specifically target these three neurons per segment based on their combinatorial expression of *Per*, *Gad1*, and *TJ*. In this scheme *Per-Gal4* drives the expression of the LexA DNA Binding Domain (*UAS-LexA^DBD*) while *Gad1* ensures the expression of the Activation Domain (*Gad1^AD*), thus allowing the reconstitution of a functional LexA protein. The flippase source is provided by *TJ-Flp*. A large proportion (~70%) of these triple intersectional larvae, in which three TJ[+]/*Per*[+] GABAergic INs per segment have been manipulated, displayed a reduction in their number of peristaltic waves upon neuronal activation compared to control larvae at 31 °C (Fig. 8a; compare Supplementary Movies 19 and 20), while a minority (~30%) appeared unaffected. Stochastic or low dTrpA1 expression in some of the neurons could explain the heterogeneity within the experimental group. In support of such a possibility, we noticed rather weak *Per-Gal4* expression in these neurons, which would be predicted to reduce the levels of the LexA^DBD component in this triple intersectional approach. Automatized tracking of the larvae showed a significant reduction of the speed of locomotion (Fig. 8d), while other larval behavior patterns, including the average body angle, appeared normal (Fig. 8b, c, e, Supplementary Movie 21), indicating that these three neurons per segment modulate crawling speed but do not impinge on turning or head casting behavior of the larvae.

Together these results show that activation of a population of three GABAergic TJ[+]/*Per*[+] neurons per segment impacts the speed of locomotion in *Drosophila* larva.

**A unique TF code in TJ[+] GABAergic neurons**. Given the medial position of the three GABAergic TJ[+]/*Per*[+] neurons and the fact that they appear to be unpaired, we hypothesized that these neurons are a subset of the midline cells. Midline cells belong to the *single-minded* (*sim*) domain[31]. We confirmed that the TJ[+] median GABAergic neurons are indeed *sim*[+] by quadruple immunostaining late embryonic stage 17 embryos with TJ, Gad1, Prospero (Pros), and *sim-Gal4* driving a nuclear GFP (Fig. 9a, b; single and double empty arrowheads). We noticed weak *sim-Gal4*-driven expression in these neurons, an observation consistent with low *sim* expression in late embryonic stages as previously reported[32]. It is important to note that the ventral TJ[+] non-GABAergic (glutamatergic-positive) neurons are not *sim-Gal4*[+] (depicted by arrows in Fig. 9c). We next found that two of the three TJ[+] GABAergic midline cells belong to the median neuroblast (MNB) progeny subpopulation identified by nuclear Pros expression (Pros-nucl)[31] (Fig. 9a, double empty arrowheads). We also found that all three TJ[+] GABAergic cells are *forkhead*[+] (*fkh*[+]) and Engrailed[+] (En[+]) (Fig. 9d, e; full arrowheads), two TFs known to be expressed in a subpopulation of MNB progeny, but also iVUMs[31]. To further delineate the exact identity of the third TJ[+] GABAergic midline neuron (*Per*[+], *fkh*[+], En[+], Pros[−]) we examined the stage 16 embryo in which

midline cell identities can be determined by their highly stereotyped dorso-ventral and anterior–posterior locations. Using these stereotyped positions along with *Per*, *fkh*, and TJ immunostainings, we showed that TJ is not expressed in the iVUMs (Supplementary Fig. 6c, f; asterisks), which are easily recognizable by their ventral-most position among midline cells, located close to the posterior boundary of the segment and posterior to H-cell sib (Supplementary Fig. 6b, e; empty arrowhead). Instead, we found TJ expression in cells located above the iVUMs in a position where the MNB progeny neurons are normally located (Supplementary Fig. 6b, c, e; circled cells and full arrowhead). We conclude that the three ventral TJ[+] GABAergic INs are MNB progeny neurons.

A class of GABAergic INs in the vertebrate spinal cord, the cerebrospinal fluid-contacting neurons (CSF-cNs, also known in Zebrafish as KA for Kolmer–Agduhr neurons), have been shown to regulate the speed of locomotion[3,8]. This class of spinal INs shares several features with the well-characterized V2b IN subpopulation. Both are GABAergic and can be distinguished based on the combinatorial expression of highly specific TFs: *Forkhead Box A2* (*Foxa2*), GATA-binding factor 2/3 (*Gata2/3*), and *Stem cell leukemia/T-cell acute leukemia 1* (*Scl/Tal1*). To determine whether the molecular identity of Drosophila Gad1[+] MNB progeny neurons could be similar to the CSF-cNs and V2b INs, we analyzed the expression of the respective *Drosophila* orthologues *Fkh*, *Grain*, and *Helix loop helix protein 3b* (*Hlh3b*). We found that the MNB progeny neurons express all three orthologues (*Hlh3b* shown in Fig. 9h, i; full arrowheads; *Fkh* and *Grain* shown in Fig. 9k–m; full arrowheads). MNB progeny neurons arise from MNB neuroblast; strikingly, at stage 15 we found that this neuroblast expresses the TF Jumeau (Jumu), the *Drosophila* orthologue of vertebrate *Foxn4*, which is expressed by the progenitors of the CSF-cNs (Fig. 9g; double full arrowhead).

The morphologies of MNB progeny neurons has been only sparsely reported[31,33]. To visualize the anatomy of single MNB progeny neurons, we took advantage of a lexA driver from the Janelia Research Campus Flylight collection, *R47G08-lexA*, whose expression we found to co-localize with TJ specifically in one to two of the MNB progeny neurons per segment. Used in combination with *TJ-Flp*, *R47G08-lexA* allowed us to visualize the morphology of single TJ[+] MNB progeny neurons. These neurons have cell bodies located ventrally, and each sends a single neurite dorsally that then bifurcates and extends both anteriorly and posteriorly in the longitudinal connectives (Fig. 10a–c). This morphology is strikingly similar to that of the previously described GABAergic Ladder neurons[17]. Indeed, using a split Gal4 line that selectively targets Ladder-d neurons[17] we found that TJ is expressed in this subtype (Fig. 10d–g, Supplementary Movie 22).

In summary, we identified the TJ[+] *Per*[+] GABAergic neurons that modulate locomotor speed as three MNB progeny neurons

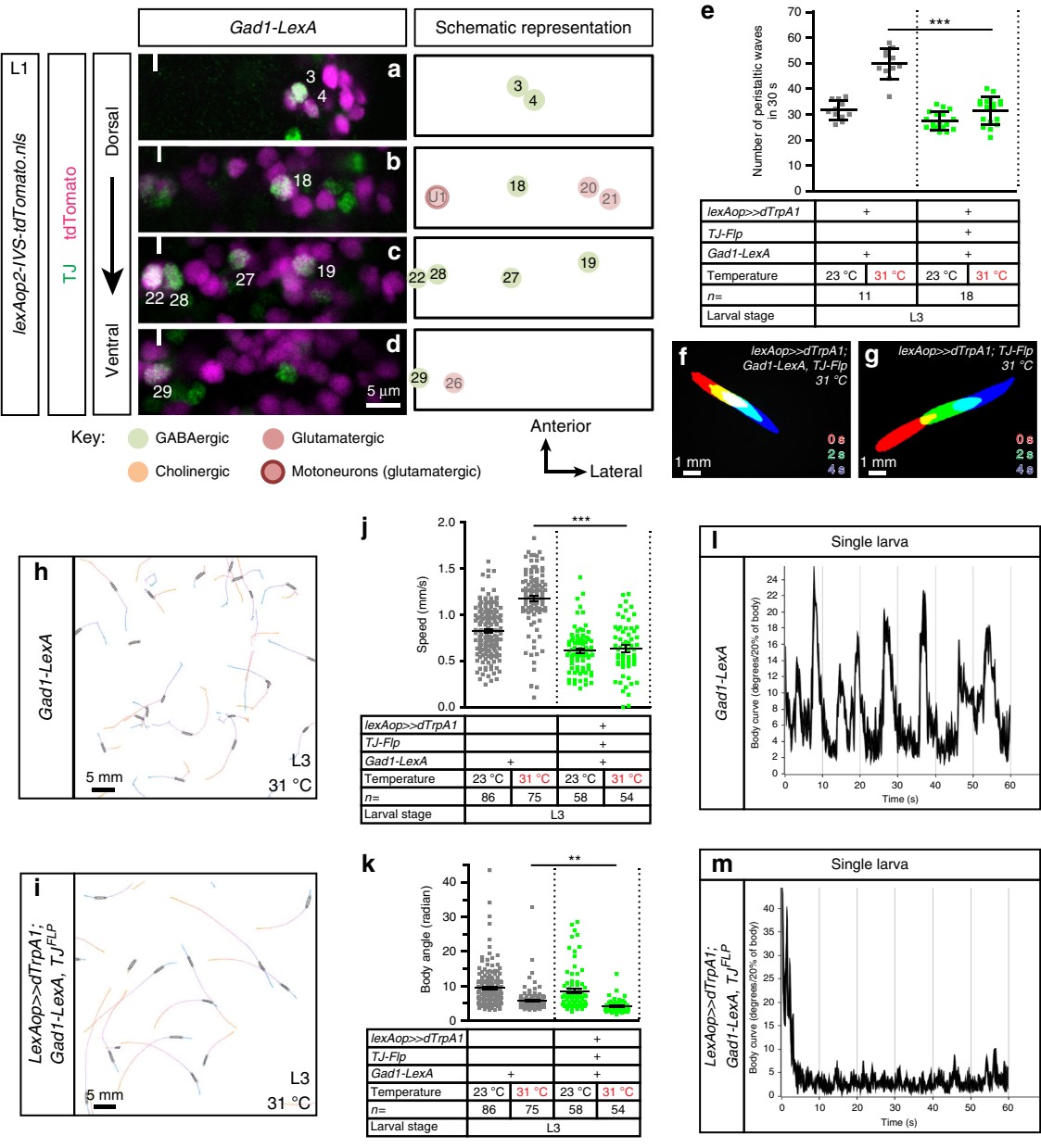

**Fig. 6** Activation of TJ[+] GABAergic INs reduces crawling speed of larvae navigating in smooth runs. **a–d** Representative views of one hemisegment of a first instar larval VNC stained for TJ (green) and GABAergic cells (magenta, using *Gad1-LexA*-driving *lexAop-nlsTomato*). Cells are shown from dorsal (**a**) to ventral (**d**) positions. The full line indicates the midline and anterior of the VNC is up. TJ[+] cholinergic GABAergic neurons are represented in bright color while other non-GABAergic neurons are shown with paler colors. The identity of those neurons is inferred from their position compared to TJ[+] GABAergic neurons. **e** Number of peristaltic waves per 30 s at (23 °C) and (31 °C) of larvae expressing TrpA1 in TJ[+] GABAergic neurons (green dot plot) versus their respective controls that do not express TrpA1 (gray dot plot). Each single point represents a single third instar larva. Error bars indicate the mean ± SD and *n* the number of larvae tested. Statistical analysis: one-way ANOVA. ***$p \leq 0.001$. **f, g** Superimposition of three consecutive time frames (0, 2, and 4 s) showing the postures of control third instar larvae (**f**) and upon activation of the TJ[+] GABAergic population (**g**). **h, i** Traces of control larvae (**h**) and upon activation of the TJ[+] GABAergic population (**i**) recorded using the MWT automated behavioral device. **j** Speed of larvae expressing TrpA1 in TJ[+] GABAergic neurons (green dot plot) versus their respective controls that do not express TrpA1 (gray dot plot). **k** Body angle (in radian) of larvae expressing TrpA1 in TJ[+] GABAergic neurons (green dot plot) versus their respective controls that do not express TrpA1 (gray dot plot). **j, k** Statistical analysis: Z-test. ***$p \leq 0.001$. **$p \leq 0.01$. **l, m** Body curve measurement for a single control larva (**l**) and a single larva upon activation of the TJ[+] GABAergic population (**m**)

derived from a Jumu[+] progenitor and uniquely defined by the combinatorial expression of TJ, En, *fkh*, *Per*, *Hlh3b,* and *Grain*.

## Discussion

In this study we characterized the neurons within the *Drosophila* VNC expressing the evolutionarily conserved TF TJ and

investigated their role in generating crawling behavior of freely moving larvae. We generated a *TJ-Flp* line and developed an intersectional genetic approach based on the expression of TrpA1 to specifically activate different TJ[+] neuronal subpopulations defined by their neurotransmitter properties.

Our time course analysis revealed that TJ is expressed in the same restricted subset of neurons from their birth in the embryo

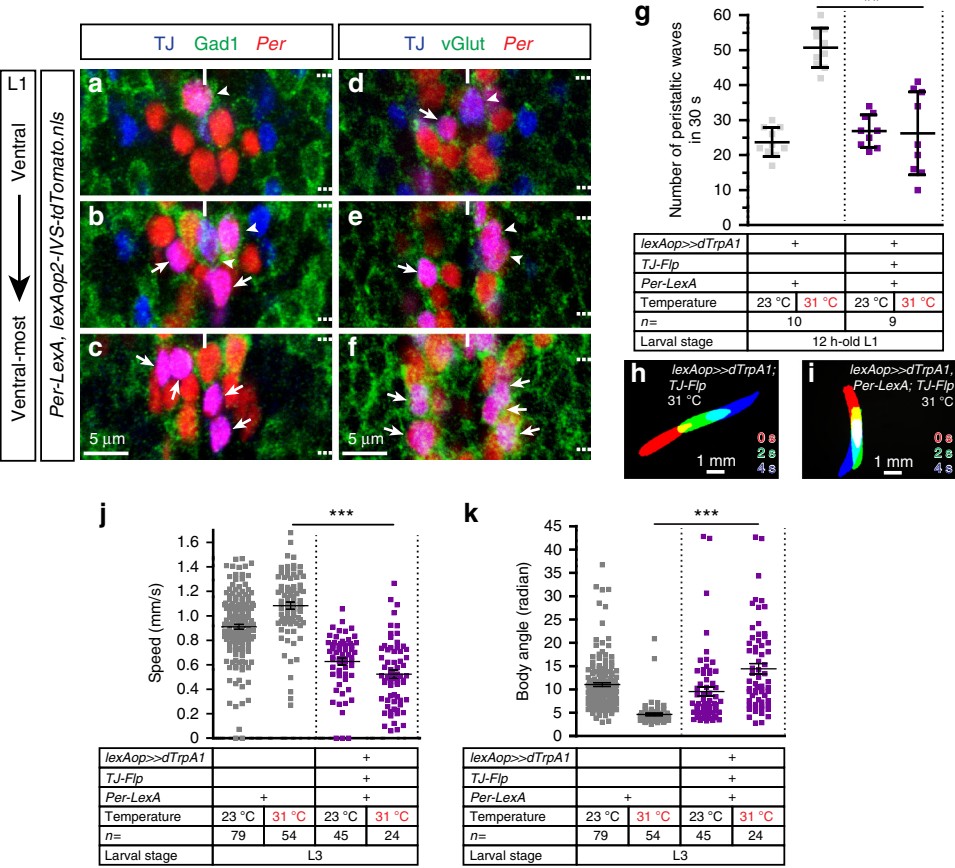

**Fig. 7** Activation of TJ⁺/*Per*⁺ INs reduces crawling speed and triggers active head casts. **a–c** Staining for TJ (blue), Gad1 (green), and *nls-tdTomato* (red) driven by *Per-LexA* in freshly hatched first instar larva VNC. Staining reveals nine TJ⁺/*Per*⁺ cells per segment, located in the ventral part of the VNC. The three most dorsal cells, located at the midline, are GABAergic (full arrowheads), while the six other cells are not (arrows). Note that one of the TJ⁺/*Per*⁺ GABAergic neurons (often the most dorsal) is weakly *Per*⁺. **d–f** Staining for TJ (blue), vGlut (green), and *nls-tdTomato* (red) driven by *Per-LexA* in freshly hatched first instar VNC. Six TJ⁺ glutamatergic cells per segment can be visualized (arrows) while the three other TJ⁺/*Per*⁺ cells are not (full arrowheads, presumably GABAergic). **g** Number of peristaltic waves per 30 s at (23 °C) and (31 °C) done by larvae expressing TrpA1 in TJ⁺/*Per*⁺ neurons (purple dot plot) versus controls that do not express TrpA1 (gray dot plot). Activation of the TJ⁺/*Per*⁺ population (nine cells per segments) leads to a decrease in the speed of locomotion of 12-h-old larvae (purple dot plot). Error bars indicate the mean ± SD and *n* the number of larvae tested. Statistical analysis: Kruskall–Wallis **$p \leq 0.01$. **h, i** Superimposition of three consecutive time frames (0, 2, and 4 s) showing the postures of third instar larvae: control larva (**h**) and upon activation of the TJ⁺/*Per*⁺ population (**i**). **j** Speed of larvae expressing TrpA1 in TJ⁺/*Per*⁺ neurons (purple dot plot) versus their respective controls that do not express TrpA1 (gray dot plot). **k** Body angle (in radian) of larvae expressing TrpA1 in TJ⁺/*Per*⁺ neurons (purple dot plot) versus their respective controls that do not express TrpA1 (gray dot plot). **j, k** Statistical analysis: *Z*-test. ***$p \leq 0.001$

through the L3 larval stage. From our comprehensive mapping we found that the number of TJ⁺ neurons (29 neurons/hemisegment) remains constant within the abdominal A2–A6 region of the VNC throughout embryonic and larval development. Based on their neurotransmitter usage, the TJ⁺ neurons can be further subdivided into three subpopulations: 10 TJ⁺ cholinergic INs, 11 TJ⁺ glutamatergic neurons, and 8 TJ⁺ GABAergic INs. Activation of the TJ⁺ cholinergic subpopulation gives rise to a ventral contraction phenotype, with larvae frequently adopting a "crescent shape" position and occasionally rolling. We noticed that the ventral contraction phenotype is heterogeneous between individuals. Those larvae with the most dramatic features were persistently immobile and ventrally curved, but peristaltic waves could still be observed emerging from the posterior region of the body. Another group of larvae displayed bouts of ventral contraction interrupting otherwise seemingly normal crawling phases that were characterized by regular propagation of peristaltic waves along the body. Recently, Clark and collaborators (2016)[9], while surveying the Janelia collection of Gal4 lines[34] crossed to *UAS-TrpA1*, identified several lines that gave rise to

similar ventral contraction phenotypes. Interestingly, they also reported a spectrum of severity: in some larvae, crawling was continually blocked by tonic contraction of ventral muscles, while in others there would be periods of ventral contraction followed by attempts to crawl. From this screen, three different lines giving rise to these phenotypes showed expression in subsets of INs. It will be of interest to determine in future studies if these subsets include TJ⁺ cholinergic INs. Interestingly, a specific population of premotor cholinergic INs in the VNC, named Down-and-Back (DnB) neurons, have recently been found to promote nociceptive crescent shape and rolling escape response[35]. In light of these observations we currently favor the hypothesis that TJ⁺ cholinergic INs are part of a neuronal circuit controlling the "crescent shape" and/or rolling behavior of larvae whereby either they trigger one or the other behavior probabilistically, or alternatively the "crescent shape" could indicate an unsuccessful or uncompleted attempt to roll. It will therefore be interesting to investigate whether there is overlap between the DnB neurons and TJ⁺ cholinergic INs identified in the present study.

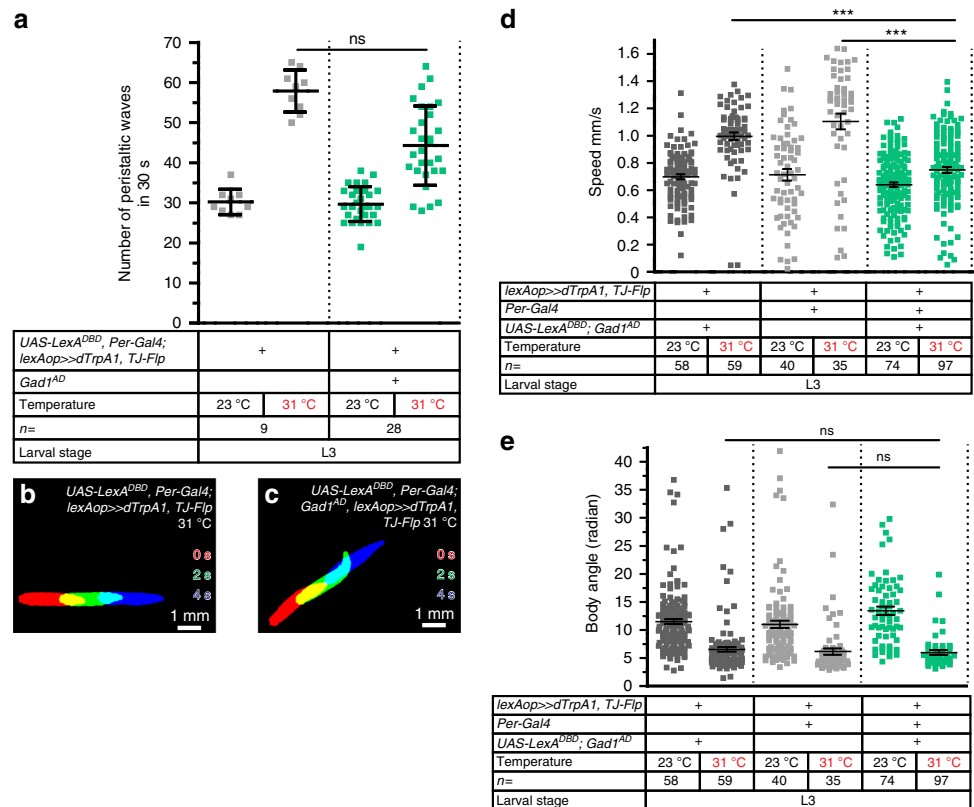

**Fig. 8** Three TJ$^+$/*Per*$^+$ GABAergic neurons per segment impact locomotor speed. **a** Number of peristaltic waves per 30 s at (23 °C) and (31 °C) temperatures done by larvae expressing TrpA1 in three TJ$^+$/*Per*$^+$ GABAergic neurons per segment (green–blue dot plot) versus controls that do not express TrpA1 (gray dot plot). Activation of the TJ$^+$/*Per*$^+$ GABAergic population (three cells per segments) leads to a decrease (albeit non-significant) in the number of peristaltic waves done by third instar larvae (second green–blue dot plot). Error bars indicate the mean ± SD and *n* the number of larvae tested. Statistical test: Kruskall–Wallis. ns = non-significant. **b**, **c** Superimposition of three consecutive time frames (0, 2, and 4 s) showing the postures of third instar control larvae (**b**) and upon activation of the TJ$^+$/*Per*$^+$ GABAergic population (**c**). **d** Speed of larvae expressing TrpA1 in three TJ$^+$/*Per*$^+$ GABAergic neurons (green–blue dot plot) versus their respective controls that do not express TrpA1 (gray dot plot). **e** Body angle (in radian) of larvae expressing TrpA1 in three TJ$^+$/*Per*$^+$ GABAergic neurons (green–blue dot plot) versus their respective controls that do not express TrpA1. **d**, **e** Statistical analysis: *Z*-test. ***$p \leq 0.001$. **$p \leq 0.01$. ns = non-significant

Activation of TJ$^+$ GABAergic neurons gives rise to slowed locomotion. The automated monitoring of larval behavior revealed that the larvae perform straight and smooth runs upon activation of these neurons. This may represent an active suppression of turning, since one would expect that as crawling speed is decreased larvae would have greater opportunity to turn. Further subdivision within the TJ$^+$ GABAergic IN pool using a triple intersectional genetics approach revealed that three TJ$^+$/*Per*$^+$ GABAergic INs per segment located at the midline and known as MNB progeny neurons substantially impact the crawling speed of the larvae. It thus appears that this *Per*$^+$/GABAergic$^+$ population was overlooked in the previous characterization of the PMSIs. This is likely due to the fact that of the 20 *Per*$^+$ INs present in each segment, the three GABAergic INs express low levels of *Per*, especially in third instar larvae, the stage in which the PMSIs were originally characterized[5].

Cell body position and molecular characterization of the TJ$^+$/*Per*$^+$ GABAergic INs allowed us to identify them as a subset of the MNB progeny. Although development of the midline cells has been particularly well described[31–33], the functional role of these cells in locomotion or other behaviors is poorly understood[36,37]. Here we show for the first time that MNB progeny neurons have a relevant function in larval locomotor behavior. Our detailed molecular characterization of the TJ$^+$/*Per*$^+$ GABAergic MNB progeny INs allowed us to survey for potential counterparts in other model species. We found that the TJ$^+$/*Per*$^+$ GABAergic INs are defined by their expression of *Hlh3b* and *Grain* (see summary schematic, Fig. 9o). In the mouse, the respective orthologous genes are *Tal1/Tal2/Scl* and *Gata2/3*, a TF code specifically found in V2b GABAergic INs[16]. Interestingly, the V2b IN subpopulation is known to regulate, in cooperation with V1Ia INs, the limb CPG coordinating flexor–extensor motor activity[38]. V2b INs derive from a *Foxn4*$^+$ progenitor domain[39]; we find similarly that the MNB neuroblast which gives rise to the MNB progeny neurons expresses *Jumu*, the *Drosophila* ortholog of *Foxn4*. Additionally, TJ$^+$/*Per*$^+$ GABAergic MNB progeny INs express *Fkh* and are midline cells derived from *sim*-expressing precursor cells. Intriguingly, in the mouse, a subset of CSF-cNs express *Foxa2*, the vertebrate orthologue gene of *Fkh* and arise from the *Sim1*$^+$ progenitor domain p3 bordering the floor plate (see summary schematic, Fig. 9p). This subset denoted CSF-cN″ is characterized by their very ventral location in the spinal cord abutting the floor plate[40]. Since GABAergic$^+$ CSF-cN INs have been reported to modulate slow locomotion and body posture in zebrafish[3,8] it is tempting to speculate that *sim*/Jumu-derived TJ$^+$/*Per*$^+$/*Fkh*$^+$ MNB progeny neurons are the *Drosophila* counterparts of vertebrate *Sim1*-derived CSF-cN″ and *Foxn4*-derived CSF-cN′ neurons (see cartoon Fig. 9p). We also found that MNB progeny neurons belong to the recently characterized "Ladder" neurons, a subtype of feedforward inhibitory INs that mediate behavioral choice during mechanosensory responses in the larva[17]. Thus, our present results suggest that these neurons have an additional role during ongoing

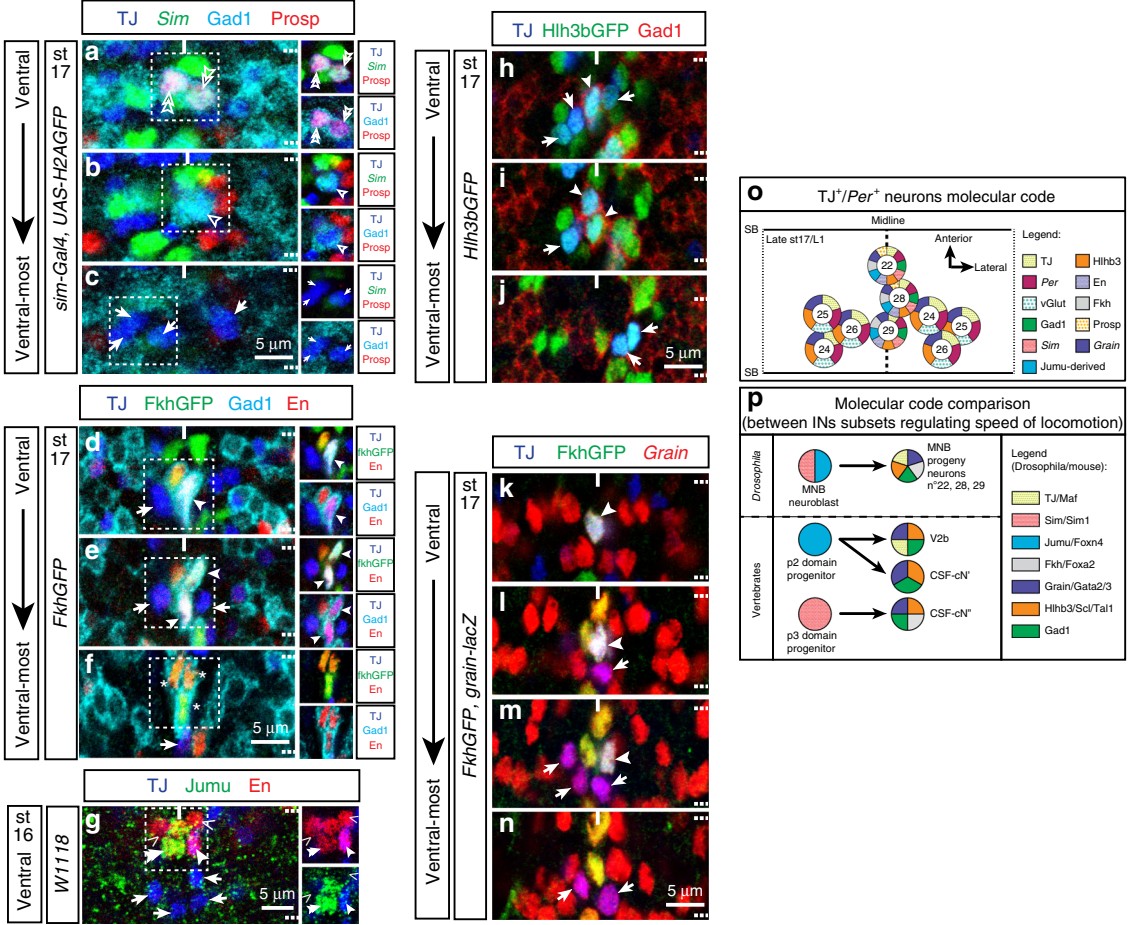

**Fig. 9** Molecular characterization of the three TJ+/Per+ GABAergic neurons. **a–c** Staining for TJ (blue), Gad1 (cyan), Prospero (Prosp, red), and GFP (green) driven by *sim-Gal4* in late stage 17 embryonic VNC. **a**, **b** *sim-Gal4* (green) is weakly expressed in ventral TJ+ GABAergic+ neurons, identifying them as midline cells (simple and double empty arrowheads). **c** TJ+ GABAergic− ventral-most neurons (which are TJ+ glutamatergic neurons, pointed by arrows) are *sim-Gal4*−, hence not part of the midline cells. Among the three TJ+/*sim+weak*/GABAergic+ located at the midline, two are positive for Prospero (**a** double empty arrowheads) and one negative (**b** simple empty arrowhead). **d–f** Staining for TJ (blue), Gad1 (cyan), Engrailed (En, red), and FkhGFP in stage 17 embryonic VNC. Ventrally, TJ+/GABAergic+ neurons (full arrowheads) are En+ and Fkh+, two markers of MNB progeny neurons. Note in the ventral-most part of the VNC the three iVUMs cells En+/Fkh+/Gad1+ that are TJ− (asterisks in panel **f**). **g** Staining for TJ (blue), Jumu (green), and En (red) in stage 16 embryonic VNC. The MNB neuroblast, identified by its position and size, is Jumu+ (double full arrowhead). Note the presence of En+/TJ− (chevrons) and En+/TJ+ MNB progeny neuron (full arrowhead) located in close proximity to the MNB neuroblast. Some TJ+/En− cells located away from the MNB neuroblast can be visualized (arrows). **h–j** Staining for TJ (blue), Gad1 (red), and Hlh3bGFP (green) in late stage 17 embryo VNC. Both TJ+/GABAergic+ (full arrowhead) and TJ+/GABAergic− (arrows) are Hlh3b+. **k–n** Staining for TJ (blue), FkhGFP (green), and βgal (red) from *Grain-lacZ* in stage 17 embryonic VNC. Both TJ+/GABAergic+ (full arrowhead) and TJ+/GABAergic− (arrows) neurons are Grain+. **o** Schematic representation of the molecular code for TJ+/Per+ neurons in late stage 17 embryo/L1 larva. A full segment is shown; SB indicates segment boundary. **p** Comparison of the molecular codes found in TJ+ MNB progeny neurons and the MNB neuroblast with the code found in the V2b interneurons, CSF-cNs and their progenitors. Transcription factor codes in vertebrates is based on the recent work of Andrzejczuk et al.[44] and Petracca et al.[40]

movements, coupling mechanosensory chordotonal neurons input with the speed of locomotion. Previous work has provided evidence for a putative circuit through which these neurons can modulate locomotor output[41]: mechanosensory input, relayed by chordotonal neurons, activates TJ+/Gad1+ MNB progeny/Ladder-d neurons, which through inhibition of so-called Basin neurons can modify motor output either through projections to the brain, or directly through connections with premotor circuitry (Fig. 10h). This observation further extends the analogy between the Ladder neurons and the CSF-cN in zebrafish. Indeed, in each system a GABAergic population of INs that share a highly similar TFs profile contributes to the control of speed via a mechanosensory circuit[42]. The powerful connectomics toolkit of the *Drosophila* larva will now allow us to dissect the neural circuitry involved in this process in this animal. A recent study focusing on the

molecular characterization of neuronal types in the annelid *Platynereis dumerilii* raises the possibility that a group of neurons specifically co-expressing the TFs *Gata1/2/3* and *Tal* are related to the CSF-cN neurons[43], suggesting that the molecular nature and physiological function of this neuronal type have been conserved during evolution between annelids, arthropods, and chordates. The remarkable similarities of TF combinatorial expression within this IN class further argues that the molecular mechanisms used during the wiring of the locomotor system are conserved and evolutionarily ancient.

## Methods

**Construction of *TJ-Flp*.** We chose to use pFly-Fos technology[23] to generate *TJ-Flp* line. Ejsmont et al.[23] generated among their fosmid library a pFly-TJ-fosmid that contains the full *tj* sequence and probably all of *tj* regulatory elements. Briefly, in

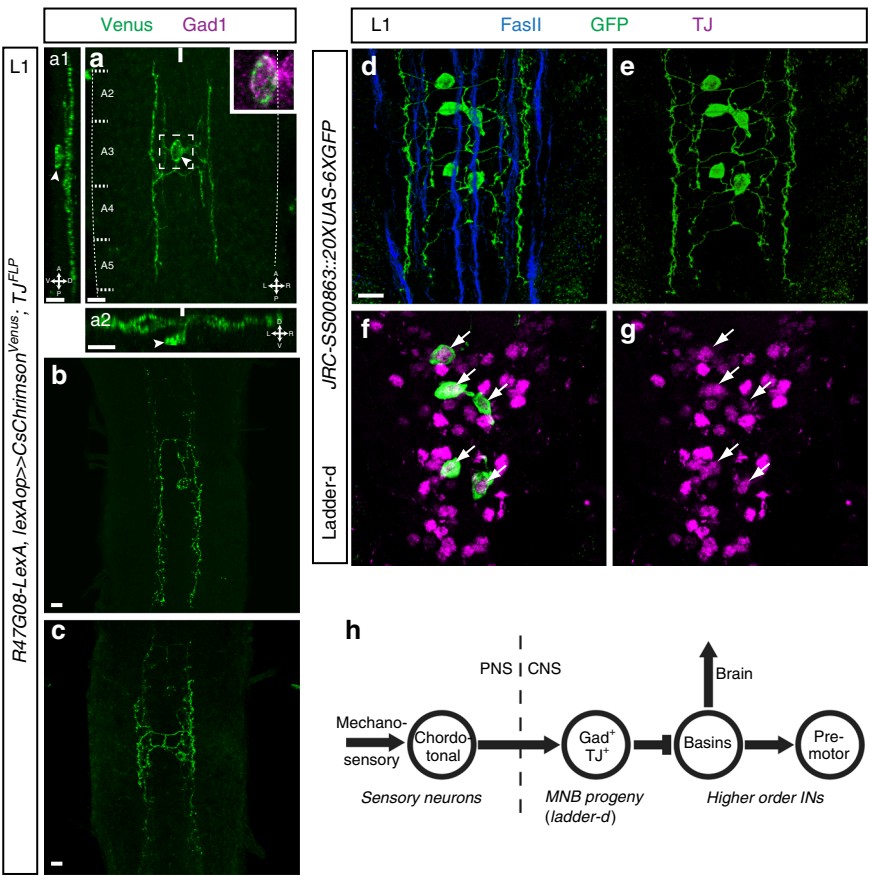

**Fig. 10** The TJ⁺ MNB neurons are Ladder neurons. **a** 3D reconstruction of a first instar larva VNC stained for Gad1 (magenta) and showing the morphology of a single GABAergic TJ⁺ MNB progeny neuron (arrowhead, expressing Venus under the combined expression of *TJ-Flp* and *R47G08-lexA*) in dorsal view (**a**), lateral view (**a1**), and posterior view (**a2**). For all three views (**a**, **a1**, **a2**), the magenta channel (Gad1 staining) was omitted to allow a better observation of the MNB progeny neuron morphology. Close-up in **a** confirms that the TJ⁺ MNB progeny neuron shown is indeed GABAergic. **b**, **c** Dorsal projection of first instar larva single MNB progeny neurons. **d–g** 3D reconstruction of a first instar larva VNC showing the morphology of Ladder-d neurons visualized using the split Gal4 line *JRC-SS00863* driving *20XUAS-6XGFP* and stained for FasII and TJ. **f**, **g** Single confocal sections through the soma showing that Ladder-d neurons express TJ (arrows). **h** Proposed circuit for TJ⁺/Gad1⁺ MNB progeny-dependent modulation of the speed of locomotion. Mechanosensory input, relayed by chordotonal neurons, activates TJ⁺/Gad1⁺ MNB progeny (Ladder-d) neurons, which through inhibition of Basin neurons (Basins) can modify motor output either through projections into the brain or directly through connections with premotor circuitry. Scale bars: 5 µm

this fosmid and following Ejsmont and collaborators (2009) approach and methodology, we replaced *tj* open reading frame and 3′ UTR by an optimized flippase sequence followed by a late SV40 polyadenylation sequence and a kanamycine resistance gene using the recombination oligonucleotides recTJsens and recTJantisens listed in the above table. Recombination oligonucleotides were chosen in order to: (i) conserve the two endogenous *tj* transcription start signals (ATG) upstream of the flippase sequence in the *TJ-Flp* line and (ii) conserve *tj* polyA signal sequence. Following recombination and contrary to Ejsmont protocol, we did not remove the kanamycine resistance sequence from the final fosmid used for transgenesis. Final fosmid was sent for targeted transgenesis in attp2 landing site (IIIrd chromosome) (Bestgene Inc.).

**Immunohistochemistry**. Briefly, larvae of the desired genotype were sorted out in Phosphate Buffered Saline (DPBS 1× CaCl2⁺ MagCl2⁺ Gibco Invitrogen, Sigma-Aldrich) + 0.1% triton (Triton X-100; Sigma Life Science) (PBST), rinsed in DPBS, and transferred with tweezers to the DPBS-filled dissection chamber. Dissection chamber consists of a silicone-delineated well on poly-lysine-coated glass slides; a double-sided piece of adhesive tape covers part of the dissection chamber floor. Dissection takes place on the adhesive tape. The head of the larvae was cut with scissors and posterior part of the body removed. Central nervous systems (CNS) were delicately dissected with tweezers (second and third instar larvae) or tungsten wires (first instar larvae) and all other tissues removed. CNS were then transferred to the non-adhesive covered part of the chamber and left to adhere to the slide, ventral part of the VNC against the slide (apart for period-observing dissections: CNS were placed dorsal part of the VNC against the slide). Further incubation steps were completed in the silicone chamber. CNS were fixed for 15 min with 3.7% formaldehyde diluted in DPBS (37% formaldehyde solution, ref 252549; Sigma-Aldrich), then washed 3 × 5 min with PBST. At this point the double-sided piece of

tape was removed. Blocking step was performed with PBST supplemented with 4% donkey serum (normal donkey serum S30–100 ml; Millipore) and 0.02% azide (Sodium azine NaN₃, ref S2002; Sigma-Aldrich) for 30 min. CNS were then incubated with primary antibodies for 1 h at room temperature (RT) or overnight at 4 °C in a humid chamber, washed 3 × 5 min with PBST, and then incubated with secondary antibodies 1 h at RT in a humid chamber in darkness. After 4 × 5 min of washing with PBST, silicone walls of the chamber were scraped off and Mowiol and a coverslip placed over stained CNS for imaging. For first instar larval (stL1) dissection embryos were first preselected during the time when their main dorsal tracheae begin to fill with air, which represents 18 h after egg laying (AEL) and allowed to develop for further 3 h. Third instar larvae were heat-killed at 56 °C for 5 s before dissection.

**Image acquisition and processing**. Images were acquired on a Zeiss LSM700 confocal with ×40 or ×63 objectives, treated and cropped in Photoshop (Adobe) and assembled in the Illustrator (Adobe). For the benefit of color-blind readers, double-labeled images were falsely colored in Photoshop. 3D projection of whole VNC was implemented using the Zen (Zeiss software). The 3D in depth color code for Supplementary Fig. 1D4 was obtained using the ImageJ software and a macro developed by Kota Miura.

**Locomotion assay**. Locomotion was assessed through two methods: the number of peristaltic waves was manually assessed, while speed, body curve, and crab speed measurements were obtained using a specific tracking software previously described[17,28]. For the manual assessment of the number of peristaltic waves in L1 larvae hatching time of the larvae was carefully monitored and larvae locomotion was assessed 6 h after hatching. For third instar larvae experiments,

eggs were laid for 5 h on basic maize feeding medium. Approximately 72 h later burrowing third instar larvae were picked up and assessed for locomotion. Larvae clean of food were gently picked up with tweezers (in the case of the third instar larvae) or the back of tweezers (for first instar larvae) and placed on a 56-mm-agar plate supplemented with grape juice. After a 30-s acclimation period, the number of peristaltic waves done by the larvae was manually assessed for 30 s using a binocular microscope (at that time plate surface temperature was 23 °C). The plate was transferred on a hot plate to heat for 2 min and 30 s or until it reached 31 °C. The plate was quickly removed from the heat and the number of peristaltic waves done by the larvae manually assessed for 30 s more. Plate was left to rest for 4 min until surface temperature reached 23 °C. Number of peristaltic waves in 30 s was assessed once more. All behavior assays were conducted by the same experimenter. Statistical tests were carried out using Graphpad Prism (Graphpad software, Inc.). We tested for the homogeneity of variances between datasets with a Bartlett Test. When variances were homogeneous, we used one-way ANOVA with a Tukey post hoc test to analyze more than two groups of data. When variances were significantly different, we used Kruskall–Wallis test with a Dunn's post hoc test. For each plotted graph, the type of statistical test used for the analysis can be found in the legend of the figure. For the automatic assessment of the speed, body curve, and crab speed using the MWT (Multi-Worm Tracker) the embryos were collected for 8–16 h at 25 °C with 65% humidity. Larvae were raised at 25 °C with normal cornmeal food. Foraging 3 instar larvae were used (larvae reared 72–84 h or for 3 days at 25 °C). Before the experiments, larvae were separated from food using 10% sucrose, scooped with a paint brush into a sieve, and washed with water. This is because sucrose is denser than water, and larvae quickly float up in sucrose making scooping them out from food a lot faster and easier. This method is especially useful for experiments with large number of animals. We have controlled for the effect and have seen no difference in the behavior between larvae scooped with sucrose and larvae scooped directly from the food plate with a forceps. The larvae were then dried and spread on the agar starting from the center of the arena. The substrate for behavioral experiments is a 3.5% agar gel in a 256.25 cm² square plastic dishes. Larvae were kept at RT and the temperature on the rig inside the enclosure was set to 30 °C. Larvae were monitored for 60 s with MWT (http://sourceforge.net/projects/mwt) at RT. Then the agar plates covered with a lid and containing the same specimen was heated in a water bath (set up at 55 °C) in order to reach 31 °C at the surface of the 3.5% agar gel to activate TrpA1. We then immediately monitored the larvae for 60 s. This methodology allowed us to compare the behavior of the same set of larvae that were first monitored at RT and then at 31 °C. If larvae were dispersed on the agar gel we would gently move them in the center before warming up the plate in the water bath. All experiments were repeated two or three times and a total of minimum 30 larvae per genotypes was monitored. Larvae were tracked in real-time using the MWT software[29]. We rejected objects that were immobile or moved less than one body length of the larva. For each larva, MWT returns a contour, spine, and center of mass as a function of time. From the MWT tracking data we computed the key parameters of larval motion, using specific choreography variables (part of the MWT software package[28]). Specifically we computed the average speed of larvae in mm per second, the average body curve (in radians as a proxy for larval head casting), and average crab speed (sideways rolling speed in mm per second) performed by the larvae during 1 min. During this time window recording the MWT software might "loose" a larva because it touches another larva but it will start a new recording after this collision event. Because the software does not allow identifying which recording tracks belong to which larva a given larva could be recorded several times. Thus, the number of tracks (visualized by dots in the graphs) is bigger than the number of larva under investigation in a given condition. We used quantile–quantile (q–q plots) to evaluate whether the data are normally distributed. We used the $Z$-test to test to compare means (for speed, body curve, and crab speed) of larvae in which we activated specific subset of neurons with the corresponding genetic control(s). When $N$ was less then 30 we used a $t$-test.

**Reporting summary**. Further information on research design is available in the Nature Research Reporting Summary linked to this article.

## Data availability
The datasets generated during and/or analyzed during the current study are available from the corresponding author on reasonable request.

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

## Acknowledgements

We thank the Developmental Studies Hybridoma Bank at the University of Iowa, the Bloomington Stock Center and the *Drosophila* Genetic Resource Center in Kyoto Institute of Technology for monoclonal antibodies and fly stocks. We would also like to thank D. Godt, S. Heidmann, M. Frasch, H. Aberle, F.R. Jackson, M. Landgraf, Y. Aso, G. Rubin, J. Simpson, G. Miesenböck, C. Doe, P. Tomancak, S. Bourane, J. Enriquez and S. Baulac for generously sharing fly lines, antibodies, and vectors. This work was funded by grants from INSERM and a 3-year Ph.D. funding from the Association Française contre les Myopathies (AFM) for H.B (Doctoral funding n°19408) and from ANR (17-CE37-0019) for T.J. Imaging analysis was carried out on the regional reference core facility (RIO) supported by the French Ministry of Scientific Research. Boukhaddaoui Hassan provided excellent technical assistance with confocal microscopy and Erwan Maury provided help to adapt a script for the MWT software.

## Author Contributions

A.G., P.C., and H.B. designed the research. H.B. carried out experiments and processed the data presented in all figures but the following ones: A.G. carried out the experiments leading to the results presented in Fig. 3a–i, Supplementary Figs. 1, 3 and 4. T.J. carried out the experiments requiring the use of the MWT device and performed behavioral analysis which leads to the results presented in Fig. 2l, Fig. 4i–n, Fig. 5i–k, Fig. 6h–m, Fig. 7j, k, Fig. 8d, e. C.S. produced preliminary data for the project under the supervision of S.Y. and J.B.T. J.V. financially supported the beginning of the project. J.E. participated to 3D reconstructions for Supplementary Fig. 1d–g and Fig. 10d–g and financially supported the review process. M.F.Z. provided expertise in TEM analysis that led to the identification of the MNB progeny neurons as a subset of Ladder neurons. A.G. and H.B. set up the figures and wrote the manuscript. P.C., J.B.T., C.S., T.J. and M.F.Z. reviewed and proofread the manuscript.

## Competing interests

The authors declare no competing interests.
