## [Peer Review File · Nature Communications]

Reviewers' Comments:

Reviewer #1:

Remarks to the Author:

Review on "A GABA-ergic Maf-expressing..." by Babski and colleagues, submitted to Nature Communications 2018

In order for *Drosophila* circuit-analyses to be translational, either modeling studies are useful (typically identifying functionally analogous circuits in *Drosophila* and vertebrates) or an identification of potentially homologous cells in vertebrates can be sought for. The authors of the present paper opted for the latter, genetically/ molecularly defining a set of interneurons which in *Drosophila* modulate locomotion, and pointing to potentially homologous cells in vertebrates.

I find this endeavor very timely, analytically powerful, and thus potentially of great interest to the readership of Nat Comm.

While I share the authors enthusiasm that potentially a cell group in vertebrates exists which shares the transcription factor profile of some of the TJ-Gal4 neurons, I wonder whether any additional experimental evidence could be obtained to back up this statement which at present remains hypothetical (P27).

Figure 1C-G: If the TJ-Gal4 driver strain were "faithfully" mapping anti-TJ staining, there should be only white cells, ie double-labeled cells, in panels 1C-G, right? To me it seems there are some green cells (TJ-Gal4 driven reporter expression but no TJ antibody staining) and some magenta cells (TJ antibody stain but no TJ-Gal4 driven reporter expression), however. Given the importance of the TJ-Gal4 driver strain as a reagent for the present analysis, I think the paper cannot be published without this being resolved.

I note that the topology of the cells #3456 in 1C does not seem to match the topology in 1C1.

P5: What are the data that make the authors suggest that stage 2 TJ-Gal4 expression is "representative" of the expression pattern throughout larval stages 1-3?

I note that the functional/ behavioral data presented in eg Figure 1H-I were obtained for stage 1. Is stage 1 behavior "representative" of other larval stages?

P5: I think it is elegant to use two thermogenetic effectors that allow the same temperature manipulation to be used for loss and gain of function. However, UAS-shits and UAS-dTrpA1 do not actually have logically opposed effects of inactivation/ activation, respectively. That is, UAS-shits does not "inactivate neurons". Rather, it silences output from chemical synapses. In other words, the neuron might well be still able to spike, but this would not generate a transmitter-based signal at the postsynapse. Mind electrical synapses are unaffected by UAS-shits.

Please indicate within the legends whether means or medians are displayed as horizontal lines (eg in Figure 1H).

Figure 2, legend: I am not sure whether the data displayed in panel H really warrant the statement all TJ+ cells are TshLexA+. If so, there should not be the two green cells, I think.

Figure 3G: Here and at related points, the authors used a more restrictive set of genetic controls as compared to eg Figure 2K. Can this be justified? From my experience with behavioral analyses, one cannot, or at least should rather not, rely on the performance of genetic controls that were not run at

the same time (in "parallel") with the experimental strains.

P14: One of the key statements of the paper is that different sets of TJ-Gal4 cells, when activated, lead to distinctly different behavioral "hallmarks". I can see no convincing statistical analyses to back this up. Merely looking at the number of peristaltic waves, which is changed more or less strongly, plus the examples in eg Figure 4 S-V, is not sufficient, I believe, to make the claim that the observed phenotypes differ in kind (rather than in strength).

In general, the behavioral/ statistical analysis is much less advanced than the genetics.

Reviewer #2:

Remarks to the Author:

This study characterized neurons expressing the Maf transcription factor Traffic Jam (TJ) in the central nervous system of *Drosophila* larvae. First, the authors identified 28 neurons expressing TJ in each hemi-segment of the ventral nerve system (VNC). Then they used available genetic tools or developed new ones to classify different groups of neurons to study the function each type in locomotion. They showed that different TJ-expressing neurons express different neurotransmitters (Cha, Glutamate or GABA), and that activating different neuronal subpopulations led to different behavior phenotypes.

The authors also characterized in detail the premotor interneurons (INs) expressing *per* (i.e., period-positive median segmental INs, previously implicated in locomotion) as well as TJ. The authors identified 5 neurons in each hemi-segment that co-express both *per* and TJ, and showed that a subset (medially positioned in the VNC) express GABA whereas the remainder (laterally positioned in the VNC) express glutamate. Furthermore, they showed that these different subtypes of premotor INs co-expressing TJ and *per* also express other transcription factors, including *sim* (single mediated gene), *flk*, *En*, *prosp*, *Hsib*, TFs, and *Hlh3b*.

The authors conclude that the pattern of expressed transcription factors is conserved between *Drosophila* premotor INs co-expressing *per* and TJ and vertebrate V2b INs, and that this pattern is critical for regulating the speed of locomotion in both fruit flies and vertebrates.

While the authors' characterization of TJ-expressing neurons, which involves the use of many different genetic tools, is highly detailed and their conclusion that the pattern of transcription factors is conserved between vertebrates and fruit flies is interesting, their claim that the function regulated by this pattern (i.e., the speed of locomotion) is also conserved is weak. This is because the authors have not clarified the mechanisms by which combinatorial transcription factor expression in each neuronal type contributes to behavior.

First, despite the increasing sophistication of behavioral assays, which can decompose behaviors into numerous categories and allow for detailed quantification of how cell type-specific manipulations can influence behavior, the authors only examine one behavioral output: number of peristaltic waves per in 30 s, as a purported index of speed of locomotion (Although in the text, the locomotion phenotypes are briefly mentioned, it should be quantified). Second, the video files provided by the authors reveal many different phenotypes that do not simply reflect a difference in speed of peristaltic movements. Third, because the analyses are not couched within a framework of neuronal circuits, the authors' conception of how the different subtypes of premotor INs co-expressing *per* and TJ is unclear.

For these reasons, until I see more detailed analyses of how the different neuronal subtypes of TJ-expressing neurons contribute functionally to circuit function and behavior, I recommend that the paper be submitted to a more specialized journal.

My minor comments are as follows.

1. Proper tests should be used to statistically analyze behavior. Is the behavioral measure (number of peristaltic waves in 30 s) normally distributed? If not, did you test for homogeneity of variance? Otherwise, you may have been justified in using a nonparametric test rather than a parametric test.
2. In Figure 6, please provide the full names for all genes, including those for *fkh+*, *EN+*, *iVUMs*, *H-sib*, *HIH3b*, *gain*, and *Jumu*.
3. In Figure 6, why did you test the expression of the specific transcription factors above? I have difficulty understanding the rationale for examining these in particular. Based on the expression pattern of transcription factors, the *TJ+/Per+* neurons are divided into three groups: Group A, consisting of cells 24, 25, 26; Group B, consisting of cells 29 and 22; and Group C, consisting of cell 30. I am a bit confused as to why cells 29 and 22 are located in the same position (Figure 6P).

Reviewer #3:

Remarks to the Author:

Review of "A GABAergic Maf-expressing interneuron subset regulates the speed of locomotion in *Drosophila*" by Babski, H., Surel, C., Yoshikawa, S, Valmier, J., Thomas, J.B., Carroll, P and Garcès, A.

This manuscript investigates neurons expressing the transcription factor Traffic Jam (TJ), which is expressed in ~30 neurons per hemisegment – or about 10% of all neurons – of the *Drosophila* ventral nerve cord. They provide expression data for TJ, which remains remarkably stable from mid-embryogenesis to late larval stages. They use thermogenetic activation and silencing of these neurons, and various subsets, to address the role of the *TJ+* neurons in larval locomotion. The really nice aspects of the manuscript include the beautiful intersectional genetics used to refine expression patterns of driver lines to subsets of the *TJ* neurons, and the observation that a subset of the *TJ* neurons (*Per+Tj+*) are similar in molecular marker expression to the *V2b* neurons of the mouse spinal cord. Nevertheless, there are a few issues that need to be resolved before publication.

Major issues

1. The work is broad but not deep. There is no focus on a specific subset of the *Tj+* neurons, to the level where we understand how they regulate locomotion. No interneuron-motor neuron GRASP or TEM analysis; no imaging of these interneurons during intact or fictive locomotion. It is frustrating to have the tools, but not push forward with the next level of analysis.
2. Figure 3 is very confusing and needs attention. It is not clear which panels are from the same embryo, but most likely the common letters = common embryo (e.g. A1,2,3 are the same prep). If so, then how can a U neuron be identified in the absence of Eve marker? How can there be an assignment of a *Tj+* neuron to the U5 identity when there is no documentation of U1-U5 in any panel? Panel D is missing, probably mislabeled as panel "I", and the legend does not match (no arrow that I can see). No statement on how a MN named DO5 is identified. All of the other figures are clear; this one is a mess.
3. The grammar is poor throughout. Please have author Dr. Thomas proof the manuscript!

Minor issues (authors choice to address to improve ms)

4. Sup Fig 1B claims *Tj* does not label *Repo+* nuclei but shows multiple white double positive cells.
5. *Tj-FLP* hits the majority of *Tj+* neurons in L1 and L2 but a different partial population each time; this is not mentioned or factored into the analysis.

6. The experiments showing Tj+ motor neurons are not causing the paralysis phenotype are unconvincing; not all MNs are analyzed. Have the authors tried a pan-MN driver like OK6 in larvae?
7. The triple intersectional genotype is not clearly explained anywhere; please give precise genotype in text or figure or figure legend.
8. page 20 concludes that the authors "identify the Tj+ ... population regulating the speed of locomotion as the 3 MNB progeny..." but I don't find any data to support this strong statement. Perhaps in Figure 5, but I could not pick it out.
9. page 24, lines 7-8 have broken, incomplete sentences.

Reviewers' comments:

Reviewer #1 (Remarks to the Author):

Review on "A GABA-ergic Maf-expressing..." by Babski and colleagues, submitted to Nature Communications 2018

In order for *Drosophila* circuit-analyses to be translational, either modeling studies are useful (typically identifying functionally analogous circuits in *Drosophila* and vertebrates) or an identification of potentially homologous cells in vertebrates can be sought for. The authors of the present paper opted for the latter, genetically/ molecularly defining a set of interneurons which in *Drosophila* modulate locomotion, and pointing to potentially homologous cells in vertebrates.

I find this endeavor very timely, analytically powerful, and thus potentially of great interest to the readership of Nat Comm.

While I share the authors enthusiasm that potentially a cell group in vertebrates exists which shares the transcription factor profile of some of the TJ-Gal4 neurons, I wonder whether any additional experimental evidence could be obtained to back up this statement which at present remains hypothetical (P27).

The transcription factor (TF) profile of TJ⁺/Per⁺/Gad1⁺ (MNB progeny neurons) is highly reminiscent to the TF codes found in V2b INs and CSF-cN (CerebroSpinal Fluid-contacting Neurons) in vertebrates. To extend the analogy further we decided to begin elucidating the neuronal circuit formed by MNB progeny neurons. We identified the Ladder neurons as a type of interneuron that displays a characteristic morphology highly reminiscent of the morphology of MNB progeny neurons (Fig.10D). Ladder neurons receive synapses from mechanosensory chordotonal neurons and mediate behavioral choice upon mechanosensory stimuli in the larva (Jovanic et al., 2016). Using a split Gal4 line that selectively targets Ladder-d neurons (Jovanic et al., 2016) we further found that TJ is expressed in this subtype (Fig.10D1-D4). This result thus further extends the analogy between the ladders and the CSF-cN in zebrafish. Indeed, in each system a GABAergic population of interneurons that share a highly similar TF profile contributes to the control of speed *via* a mechanosensory circuit (Knafo and Wyart, 2018).

1. T. Jovanic, C. M. Schneider-Mizell, M. Shao, J.-B. Masson, G. Denisov, R. D. Fetter, B. D. Mensh, J. W. Truman, A. Cardona, M. Zlatić, Competitive Disinhibition Mediates Behavioral Choice and Sequences in *Drosophila*. *Cell* 167, 858-870.e819 (2016).
2. Knafo, S. & Wyart, C. Active mechanosensory feedback during locomotion in the zebrafish spinal cord. *Curr Opin Neurobiol* 52, 48-53 (2018).

Figure 1C-G: If the TJ-Gal4 driver strain were "faithfully" mapping anti-TJ staining, there should be only white cells, ie double-labeled cells, in panels 1C-G, right? To me it seems there are some green cells (TJ-Gal4 driven reporter expression but no TJ antibody staining) and some magenta cells (TJ antibody stain but no TJ-Gal4 driven reporter expression), however. Given the importance of the TJ-Gal4 driver strain as a reagent for the present analysis, I think the paper cannot be published without this being resolved.

As the reviewer rightly noticed, overlapping of TJ antibody staining (magenta channel) and GFP driven by *TJ-Gal4* driver (GFP channel) does not reveal strictly white cells. When looking at the two separated channels (Supplementary Figure 7, top panels), we can observe that all TJ⁺ cells labeled by the TJ antibody do express the *TJ-Gal4* driver (reported by GFP). Interestingly in some cells, notably in some TJ⁺ motoneurons (cells 13, 14, 17, 15 in Fig. 1E-E1), expression of TJ as detected with the antibody is low; in comparison the Gal4 driver is strong in those cells, thus leading to an overlapping staining that is more green than magenta.

I note that the topology of the cells #3456 in 1C does not seem to match the topology in 1C1.

This has been now modified in the manuscript.

P5: What are the data that make the authors suggest that stage 2 TJ-Gal4 expression is "representative" of the expression pattern throughout larval stages 1-3?

I note that the functional/ behavioral data presented in eg Figure 1H-I were obtained for stage 1. Is stage 1 behavior "representative" of other larval stages?

The reviewer is correct, we only choose to show *TJ-Gal4* expression in stage L2 because it was "representative" of the expression of *TJ-Gal4* (and TJ) in the same number of neurons from early L1 to L3. To complement the functional/ behavioral data performed in stage L1 (Figure 1H-I) we have now provided in supplementary Fig.1D1-D4 the confocal images and 3D reconstruction of a dissected L1 VNC (carrying *TJ-Gal4::UAS-H2A-GFP*) and counterstained with GFP and TJ. In such preparations, we reproducibly counted 29 TJ⁺ neurons/hemisegment. In dissected L3 VNC, where segment boundaries are difficult to accurately delineate, we quantified Flp out events using *UAS>>TrpA1^{myc}, TJ-Gal4, lexAop-Flp* (and crossed to either *Gad1-LexA* or *ChAT-LexA* or *vGlut-LexA*) to visualize and quantify the number of TJ⁺ neurons that belong to each subtypes. Using this approach, we reproducibly visualized 8 TJ⁺/*Gad1*⁺, 10 TJ⁺/*ChAT*⁺ neurons, and 11 TJ⁺/*vGlut*⁺ per hemisegment (Supplementary Fig.5).

P5: I think it is elegant to use two thermogenetic effectors that allow the same temperature manipulation to be used for loss and gain of function. However, UAS-shits and UAS-dTrpA1 do not actually have logically opposed effects of inactivation/ activation, respectively. That is, UAS-shits does not "inactivate neurons". Rather, it silences output from chemical synapses. In other words, the neuron might well be still able to spike, but this would not generate a transmitter-based signal at the postsynapse. Mind electrical synapses are unaffected by UAS-shits.

We agree with the reviewer, we have now modified the text accordingly by replacing the word "inactivation" by "silence" or "silencing" over the whole manuscript.

Please indicate within the legends whether means or medians are displayed as horizontal lines (eg in Figure 1H).

The horizontal lines indicate means. We have incorporated this information in all the legends.

Figure 2, legend: I am not sure whether the data displayed in panel H really warrant the statement all TJ+ cells are TshLexA+. If so, there should not be the two green cells, I think.

For panel H in Figure 2 we have now provided the two separated channels to confirm that all TJ expressing neurons (green) are indeed *Tsh-LexA* positive (magenta) (Supplementary Figure 7, top panels).

Figure 3G: Here and at related points, the authors used a more restrictive set of genetic controls as compared to eg Figure 2K. Can this be justified? From my experience with behavioral analyses, one cannot, or at least should rather not, rely on the performance of genetic controls that were not run at the same time (in "parallel") with the experimental strains.

Our behavioral analyses using intersectional genetic depends on the use of 6 main LexA lines namely: *Tsh-LexA*, *CQ2-LexA*, *ChAT-LexA*, *Gad1-LexA*, *vGlut-LexA* and *Per-LexA*. For a given condition the control animals that we monitored were *LexA-(of interest)* combined with *LexAop>>dTrpA1*. These control animals were run at the same time (in "parallel") with the experimental strain. The rationale behind this was that we wanted to probe simultaneously i) the possible deleterious effect of a given LexA (note that *ChAT-LexA*, *Gad1-LexA* and *vGlut-lexA* are mimic insertions within the respective genes), ii) the possible leakiness of the *LexAop>>dTrpA1* cassette when combined with a LexA driver.

P14: One of the key statements of the paper is that different sets of TJ-Gal4 cells, when activated, lead to distinctly different behavioral "hallmarks". I can see no convincing statistical analyses to back this up. Merely looking at the number of peristaltic waves, which is changed more or less strongly, plus the examples in eg Figure 4 S-V, is not sufficient, I believe, to make the claim that the observed phenotypes differ in kind (rather than in strength).

In general, the behavioral/ statistical analysis is much less advanced than the genetics.

We agree with the reviewer and these insightful comments prompted us to complement our original behavioral analyses with a set of new data using a more sophisticated approach to measure several behavioral parameters beyond simply the number of peristaltic waves. To achieve our objectives we used a previously well-established quantitative behavioral analysis method allowing examining and quantifying several larval behavioral outputs in an automatized manner (1-2). The results of these new experiments are presented in Figures 4-8 and related videos. We think that these new analyses provide a better understanding of the particular behavioral phenotypes associated with the activation of specific subsets of TJ⁺ neurons.

- 1- T. Ohyama, T. Jovanic, G. Denisov, T. C. Dang, D. Hoffmann, R. A. Kerr, M. Zlatic, High-Throughput Analysis of Stimulus-Evoked Behaviors in *Drosophila* Larva Reveals Multiple Modality-Specific Escape Strategies. *PLOS ONE* 8, e71706 (2013).
- 2- N. A. Swierczek, A. C. Giles, C. H. Rankin, R. A. Kerr, High-throughput behavioral analysis in *C. elegans*. *Nature Methods* 8, 592 (2011).

Reviewer #2 (Remarks to the Author):

This study characterized neurons expressing the Maf transcription factor Traffic Jam (TJ) in the central nervous system of *Drosophila* larvae. First, the authors identified 28 neurons expressing TJ in each hemi-segment of the ventral nerve system (VNC). Then they used available genetic tools or developed new ones to classify different groups of neurons to study the function each type in locomotion. They showed that different TJ-expressing neurons express different neurotransmitters (Cha, Glutamate or GABA), and that activating different neuronal subpopulations led to different behavior phenotypes.

The authors also characterized in detail the premotor interneurons (INs) expressing per (i.e., period-positive median segmental INs, previously implicated in locomotion) as well as TJ. The authors identified 5 neurons in each hemi-segment that co-express both per and TJ, and showed that a subset (medially positioned in the VNC) express GABA whereas the remainder (laterally positioned in the VNC) express glutamate. Furthermore, they showed that these different subtypes of premotor INs co-expressing TJ and per also express other transcription factors, including sim (single mediated gene), flk, En, prosp, Hsib, TFs, and Hlh3b.

The authors conclude that the pattern of expressed transcription factors is conserved between *Drosophila* premotor INs co-expressing per and TJ and vertebrate V2b INs, and that this pattern is critical for regulating the speed of locomotion in both fruit flies and vertebrates.

While the authors' characterization of TJ-expressing neurons, which involves the use of many different genetic tools, is highly detailed and their conclusion that the pattern of transcription factors is conserved between vertebrates and fruit flies is interesting, their claim that the function regulated by this pattern (i.e., the speed of locomotion) is also conserved is

weak. This is because the authors have not clarified the mechanisms by which combinatorial transcription factor expression in each neuronal type contributes to behavior.

First, despite the increasing sophistication of behavioral assays, which can decompose behaviors into numerous categories and allow for detailed quantification of how cell type-specific manipulations can influence behavior, the authors only examine one behavioral output: number of peristaltic waves per in 30 s, as a purported index of speed of locomotion (Although in the text, the locomotion phenotypes are briefly mentioned, it should be quantified). Second, the video files provided by the authors reveal many different phenotypes that do not simply reflect a difference in speed of peristaltic movements.

We agree with the reviewer and as we mentioned above in response to a similar comment by Reviewer #1, we have significantly extended our original behavior analyses with a set of new data using a more sophisticated approach to measure several behavioral parameters beyond simply the number of peristaltic waves. To achieve our objectives we used a previously well-established quantitative behavioral analysis method allowing examining and quantifying several larval behavioral outputs in an automatized manner (1-2). The results of these new experiments are presented in Figures 4-8 and related videos.

1- T. Ohyama, T. Jovanic, G. Denisov, T. C. Dang, D. Hoffmann, R. A. Kerr, M. Zlatic, High-Throughput Analysis of Stimulus-Evoked Behaviors in *Drosophila* Larva Reveals Multiple Modality-Specific Escape Strategies. *PLOS ONE* 8, e71706 (2013).

2- N. A. Swierczek, A. C. Giles, C. H. Rankin, R. A. Kerr, High-throughput behavioral analysis in *C. elegans*. *Nature Methods* 8, 592 (2011).

Third, because the analyses are not couched within a framework of neuronal circuits, the authors' conception of how the different subtypes of premotor INs co-expressing per and TJ is unclear.

For these reasons, until I see more detailed analyses of how the different neuronal subtypes of TJ-expressing neurons contribute functionally to circuit function and behavior, I recommend that the paper be submitted to a more specialized journal.

In light of the reviewer's recommendation, we decided to begin elucidating the neuronal circuits formed by INs co-expressing *Per* and TJ. The $TJ^+/Per^+/vGlut^+$ subtype belong to the PMSIs (Period-positive Median Segmental Interneurons) a set of premotor interneurons that project their axons in the dorsal neuropile where they establish synaptic contacts with dendrites of motoneurons (Kohsaka et al., 2014).

Our study also revealed a second subpopulation of TJ^+/Per^+ neurons that express *Gad1*. Interestingly, we now show this subtype to be the previously reported MNB progeny neurons (see Fig.9 and Supplementary Fig.6). To begin to explore the neural connectivity of these $TJ^+/Per^+/Gad1^+$ neurons we identified the Ladder neurons as a type of interneuron that display a characteristic morphology highly reminiscent of the morphology of MNB progeny neurons (Fig.10D). Ladder neurons receive synapses from mechanosensory chordotonal neurons and mediate behavioral choice upon mechanosensory stimuli in the larva (Jovanic et al., 2016). Using a split Gal4 line that selectively targets Ladder-d neurons (Jovanic et al., 2016) we further found that TJ is expressed in this subtype (Fig.10D1-D4). Our study extends the Jovanic et al study in that we show that modulation of the activity of these 3 neurons affects speed of locomotion.

T. Jovanic, C. M. Schneider-Mizell, M. Shao, J.-B. Masson, G. Denisov, R. D. Fetter, B. D. Mensh, J. W. Truman, A. Cardona, M. Zlatic, Competitive Disinhibition Mediates Behavioral Choice and Sequences in *Drosophila*. *Cell* 167, 858-870.e819 (2016).

My minor comments are as follows.

1. Proper tests should be used to statistically analyze behavior. Is the behavioral measure (number of peristaltic waves in 30 s) normally distributed? If not, did you test for homogeneity of variance? Otherwise, you may have been justified in using a nonparametric test rather than a parametric test.

We have now included more detailed information about the statistical analysis through the manuscript.

2. In Figure 6, please provide the full names for all genes, including those for *fkh+*, *EN+*, *iVUMs*, *H-sib*, *HIH3b*, *grain*, and *Jumu*.

We have provided the full names for all the genes. On note, *H-sib* is not a gene but the name of a neuron (*H-sib* stand for *H-cell sibling*).

3. In Figure 6, why did you test the expression of the specific transcription factors above? I have difficulty understanding the rationale for examining these in particular.

The reviewer is correct; we have not clearly explained the rationale for examining this particular combination of transcription factors. Interestingly, while surveying the literature for GABAergic⁺ interneurons regulating speed of locomotion in other species we came across the CerebroSpinal Fluid-contacting Neurons (CSF-cNs, also known in zebrafish as KA for Kolmer-Agduhr neurons). This important class of interneurons have been particularly well studied in the spinal cord of vertebrates and share several features with the well characterized V2b interneuronal subpopulation (also GABAergic⁺). These interneurons express the following transcription factor code: *Foxa2*, *Gata2/3* and *Sc1/Tal1*. We thus analysed the *Drosophila* orthologue genes respectively *Fkh*, *Grain* and *Hlh3b* to ask whether the molecular identity of *Gad1*⁺ MNB progeny neurons could be to some extent reminiscent to CSF-cNs and V2b interneurons and indeed found it was the case. We have now changed the text in the manuscript (Page 26, 1st paragraph starting at line 499 in the revised version) accordingly to take in consideration this important comment.

Based on the expression pattern of transcription factors, the *TJ*⁺/*Per*⁺ neurons are divided into three groups: Group A, consisting of cells 24, 25, 26; Group B, consisting of cells 29 and 22; and Group C, consisting of cell 30.

Indeed, Figure 6 lacked clarity and so we have now modified this figure accordingly (see Fig.90). The reviewer is right, the *TJ*⁺/*Per*⁺ neurons are actually divided into three groups. Group A, consisting of cells 24, 25, 26 (*vGlut*⁺); Group B, consisting of cells 22 and 28

(*GAD1*⁺ and *Prospero*⁺) and group C, consisting of cell 29 (*GAD1*⁺ and *Prospero*⁻). Note that we mislabeled cells 29 and 30, they are in fact cells 28 and 29.

I am a bit confused as to why cells 29 and 22 are located in the same position (Figure 6P).

In fact, cells 29 and 22 (now cells 28 and 22 in Fig.9O) are not located in the same position (they are not hemisegment counterparts). The 3 MNB progeny neurons (cells 22, 28 and 29, see Fig.9O) are derived from the MNB neuroblast, they occupy midline positions and are often seen settling one above the other along the dorsal-ventral axis of the VNC. For clarity we have thus slightly modified the positions of cells 28 and 22 on the graphical cartoon (Fig.9O) to ensure that they will not be perceived as hemisegment counterpart (they are unique asymmetrical cells located at the midline).

Reviewer #3 (Remarks to the Author):

Review of “A GABAergic Maf-expressing interneuron subset regulates the speed of locomotion in *Drosophila*” by Babski, H., Surel, C., Yoshikawa, S, Valmier, J., Thomas, J.B., Carroll, P and Garcès, A.

This manuscript investigates neurons expressing the transcription factor Traffic Jam (TJ), which is expressed in ~30 neurons per hemisegment – or about 10% of all neurons – of the *Drosophila* ventral nerve cord. They provide expression data for TJ, which remains remarkably stable from mid-embryogenesis to late larval stages. They use thermogenetic activation and silencing of these neurons, and various subsets, to address the role of the TJ+ neurons in larval locomotion. The really nice aspects of the manuscript include the beautiful intersectional genetics used to refine expression patterns of driver lines to subsets of the TJ neurons, and the observation that a subset of the TJ neurons (*Per*+*Tj*+) are similar in molecular marker expression to the V2b neurons of the mouse spinal cord. Nevertheless, there are a few issues that need to be resolved before publication.

Major issues

1. The work is broad but not deep. There is no focus on a specific subset of the *Tj*+ neurons, to the level where we understand how they regulate locomotion. No interneuron-motor neuron GRASP or TEM analysis; no imaging of these interneurons during intact or fictive locomotion. It is frustrating to have the tools, but not push forward with the next level of analysis.

To begin to understand how *TJ*⁺ neurons regulate locomotion we decided to explore the framework of neuronal circuits formed by INs co-expressing *Per* and *TJ*. The *TJ*⁺/*Per*⁺/*vGlut*⁺ subtype belong to the PMSIs (Period-positive Median Segmental Interneurons) a set of premotor interneurons that project their axons in the dorsal neuropile where they establish synaptic contacts with dendrites of motoneurons (Kohsaka et al., 2014).

Our study also revealed a second subpopulation of *TJ*⁺/*Per*⁺ neurons that express *Gad1*. Interestingly, we now show this subtype to be the previously reported MNB progeny

neurons (see Fig.9 and Supplementary Fig.6). To gain insight into the neural connectivity of these $TJ^+/Per^+/Gad1^+$ interneurons we surveyed the literature for previously characterized midline located GABAergic interneurons in the *Drosophila* VNC and identified the Ladder neurons as a type of interneuron that display a characteristic morphology highly reminiscent of the morphology of MNB progeny neurons (Fig.10D). Ladder neurons receive synapses from mechanosensory chordotonal neurons and mediate behavioral choice upon mechanosensory stimuli (Jovanic et al., 2016). Using a split Gal4 line that selectively targets Ladder-d neurons (Jovanic et al., 2016) we further found that TJ is expressed in this subtype (Fig.10D1-D4). Our study extends the Jovanic et al study in that we show that modulation of the activity of these 3 neurons affects speed of locomotion.

T. Jovanic, C. M. Schneider-Mizell, M. Shao, J.-B. Masson, G. Denisov, R. D. Fetter, B. D. Mensh, J. W. Truman, A. Cardona, M. Zlatic, Competitive Disinhibition Mediates Behavioral Choice and Sequences in *Drosophila*. *Cell* 167, 858-870.e819 (2016).

2. Figure 3 is very confusing and needs attention. It is not clear which panels are from the same embryo, but most likely the common letters = common embryo (e.g. A1,2,3 are the same prep). If so, then how can a U neuron be identified in the absence of Eve marker? How can there be an assignment of a Tj+ neuron to the U5 identity when there is no documentation of U1-U5 in any panel? Panel D is missing, probably mislabeled as panel "I", and the legend does not match (no arrow that I can see). No statement on how a MN named DO5 is identified. All of the other figures are clear; this one is a mess.

We agree with the reviewer that Figure 3 lacked clarity and we have now modified this Figure accordingly. The reviewer is correct; the common letters refers to a common embryo. As such, A1,A2,A3 are different confocal views taken at different dorsal to ventral positions of the VNC from the same specimen. The same is true for B1 to B3 and C1 to C3. We have now clarified this accordingly in the figure legend with the sentence: "A1-A3 are images from a single VNC and the same applies to B1-3 and C1-3".

The positions of U motoneurons are highly stereotyped with U1 located just below the aCC/pCC pair (see panels A2 and B2 in Fig.3). Because of the counterstaining with Eve we were able to clearly visualize expression of TJ in U1, U2 and U5 (see panels B2 and B3 in Fig.3). Observation of motor axons projections in the periphery using *TJ-Gal4* also confirmed these results (see supplementary Fig.4).

The other points raised by the reviewer have been addressed.

3. The grammar is poor throughout. Please have author Dr. Thomas proof the manuscript!

We have now asked Dr Thomas to carefully proof the manuscript.

Minor issues (authors choice to address to improve ms)

4. Sup Fig 1B claims Tj does not label Repo+ nuclei but shows multiple white double positive cells.

At all stages investigated we have not found TJ expression in Repo⁺ glial cells. The reviewer is right, panel B in Sup Fig 1B is misleading. To have a good overview of TJ⁺ and Repo⁺ cells we provided in the former picture a projection view of several dorso-ventral confocal sections thus giving rise to some overlay between TJ⁺ and Repo⁺ nuclei. Nevertheless, in this image no perfect overlap between these 2 types of nuclei can be found. To avoid any misinterpretation we have now provided a new projection view obtained with a reduced number of confocal sections.

5. Tj-FLP hits the majority of Tj⁺ neurons in L1 and L2 but a different partial population each time; this is not mentioned or factored into the analysis.

The reviewer is raising a very important point. To accurately monitor the efficiency and robustness of *TJ-FLP* in time and place we decided to use a LexA/LexAop based system to complement the approach we previously used (that was based on *Act>>Gal4, UAS-CD8GFP*). We took advantage of the convenient *LexAop-FRT-stop-FRT-Chrimson-Venus* line based on the facts that i) this is a Venus-tagged construct, ii) this transgene was built from the same backbone as *LexAop-FRT-stop-FRT-dTrpA1* (used for our behavioural analysis), and iii) both transgenes (*LexAop-FRT-stop-FRT-Chrimson-Venus* and *LexAop-FRT-stop-FRT-dTrpA1*) have been inserted in the genome at the exact same location. We reasoned that *TJ-FLP* in combination with *LexAop-FRT-stop-FRT-Chrimson-Venus* will thus mimic flip out events induced by *TJ-FLP* in combination with *LexAop-FRT-stop-FRT-dTrpA1*. Using this transgene in combination with *Tsh-LexA* we found that 74,5% of the TJ-expressing neurons (n=852 TJ⁺ neurons counted) have already recombined in young L1 larvae (0-4hr old) and most if not all TJ-expressing neurons did so in late L1 larvae (>12hr old, data not shown) indicating that monitoring flip out events under this condition is particularly efficient. In the former approach (using *Act>>Gal4, UAS-CD8GFP*) an extra step is needed (i.e. production of the Gal4 protein before activation of the UAS that will trigger the production of GFP) suggesting that we may have underestimated flip out events in early specimens.

Importantly, using either approach we also observed that from early L1 specimens (0-4hr old larvae) onward flip out events are very reproducible and highly stereotyped from hemisegment to hemisegment in each VNC analyzed (see Figure 2 and Supplementary Figure 2). This observation strongly indicates that different subpopulations of TJ⁺ neurons recombine at different times rather than stochastically. Since our behavior analysis has been mostly carried out with L3 larvae this result guarantees that all TJ⁺ neurons have been efficiently recombined in these animals. Finally, since activation of a subset of TJ motoneurons using the *CQ2-lexA* has been carried out on 12hr old L1 larvae (Fig. 3G) we have carefully monitored Flip out events in these motoneurons using Eve as a marker. We have found that more than 98% of these motoneurons have been efficiently recombined in 0-4hr old larvae (n=48 hemisegment analysed, see supplementary Figure 2) thus confirming that the expected contingent of TJ⁺ motoneurons have been properly manipulated under this condition.

6. The experiments showing Tj⁺ motor neurons are not causing the paralysis phenotype are unconvincing; not all MNs are analyzed. Have the authors tried a pan-MN driver like OK6 in larvae?

As the reviewer rightly points out, ideally we would have liked to use intersectional genetics to target all TJ⁺ motoneurons. Unfortunately, we were limited by the availability of proper genetics tools to perform such an experiment. Indeed when we used *OK6-Gal4* (the widely used pan motoneuronal driver) we found expression of this driver not only in TJ⁺ motoneurons but also in a set of 3 TJ⁺ interneurons (per hemisegment). Since to our knowledge no other line is to date available allowing for specific expression in motoneurons, we decided to use the *CQ2-LexA* that allows for partial but accurate targeting of 4 of the 6 TJ⁺ MNs.

7. The triple intersectional genotype is not clearly explained anywhere; please give precise genotype in text or figure or figure legend.

We agree with the reviewer that this approach was not explained in the manuscript; we only reported the genotype of the animals bearing this triple intersection (in Figure 5 (J-L) of the previous version). An explanation of the logic of this approach and a precise description of all the transgenes used is now included in the text (Pages 21,22 from lines 416 to 421 in the revised version).

8. page 20 concludes that the authors “identify the Tj+ ... population regulating the speed of locomotion as the 3 MNB progeny...” but I don't find any data to support this strong statement. Perhaps in Figure 5, but I could not pick it out.

We have now replaced the word “regulating” by “impacting”. See pages 5 (line 85), 21 (line 402), 22 (line 433) and on title of Fig.8. Note that we also have used the following phrasing on page 33 (line 690):

“Further subdivision within the TJ+ GABAergic INs pool, using a triple intersectional genetics approach, revealed that 3 *Per+*/TJ+ GABAergic INs located at the midline and known as MNB progeny neurons substantially impact the crawling speed of the larvae”.

9. page 24, lines 7-8 have broken, incomplete sentences.

This has been now corrected.

Reviewers' Comments:

Reviewer #1:

Remarks to the Author:

Review on "A GABA-ergic Maf-expressing..." by Babski and colleagues, re-submitted to Nature Communications 2019

In assessing this re-submitted manuscript, I found it improved in particular for more detailed behavioral analyses as requested by myself and the other reviewers. However, my main reason for not recommending publication by Nature Communications remains.

That is: The authors punch line at the end of their abstract is that they define a set of interneurons that in *Drosophila* modulate locomotion, and argue that potentially homologous cells exist in vertebrates. I still do believe that this is potentially of great interest to the readership of Nat Comm. However, the authors have not significantly strengthened their point that such a cell group in vertebrates is indeed homologous. As an indication of this lack of substantial further evidence I note that there are no substantial changes to the abstract.

I am sure the present paper will fare well with a more specialized journal, and will be much appreciated by the community.

Reviewer #2:

Remarks to the Author:

The authors replied all my questions and concerns. I support this manuscript will be publish in Nature communications.

Reviewer #3:

Remarks to the Author:

This revised manuscript is much improved, and nearly all of my prior comments have been satisfactorily addressed. In particular, the addition of TEM identification of the ladder neurons as part of the TJ+ population is excellent, and allows some circuit inferences to be made.

The one point that was not properly addressed (or perhaps I'm just confused by the text) is that only a small number of TJ+ neurons (3) were used to characterize behavior. I would recommend that this is made more clear in the text: that just three neurons of the larger pool are used for the analysis.

Point-by-point response to reviewers' comments

Reviewer #1 (Remarks to the Author):

Review on "A GABA-ergic Maf-expressing..." by Babski and colleagues, re-submitted to Nature Communications 2019

In assessing this re-submitted manuscript, I found it improved in particular for more detailed behavioral analyses as requested by myself and the other reviewers. However, my main reason for not recommending publication by Nature Communications remains.

That is: The authors punch line at the end of their abstract is that they define a set of interneurons that in *Drosophila* modulate locomotion, and argue that potentially homologous cells exist in vertebrates. I still do believe that this is potentially of great interest to the readership of Nat Comm. However, the authors have not significantly strengthened their point that such a cell group in vertebrates is indeed homologous. As an indication of this lack of substantial further evidence I note that there are no substantial changes to the abstract.

I am sure the present paper will fare well with a more specialized journal, and will be much appreciated by the community.

We have now moved the sentence about cross-species relevance to vertebrate locomotion from the abstract to the discussion.

Reviewer #2 (Remarks to the Author):

The authors replied all my questions and concerns. I support this manuscript will be publish in Nature communications.

Reviewer #3 (Remarks to the Author):

This revised manuscript is much improved, and nearly all of my prior comments have been satisfactorily addressed. In particular, the addition of TEM identification of the ladder neurons as part of the TJ+ population is excellent, and allows some circuit inferences to be made.

The one point that was not properly addressed (or perhaps I'm just confused by the text) is that only a small number of TJ+ neurons (3) were used to characterize behavior. I would recommend that this is made more clear in the text: that just three neurons of the larger pool are used for the analysis.

We have now addressed this point that was not clear and followed the recommendation of the Reviewer. We have now clearly mentioned in the main body of the manuscript (top of page 35) and in the legend of Figure 8 that just three neurons were used to characterize behavior.